# Differential regulation of hair cell actin cytoskeleton mediated by SRF and MRTFB

**Ling-Yun Zhou[1,2,3], Chen-Xi Jin[1,2,3], Wen-Xiao Wang[1,2,3], Lei Song[1,2,3], Jung-Bum Shin[4], Ting-Ting Du[1,2,3]\*†, Hao Wu[1,2,3]\*†**

[1]Department of Otolaryngology-Head and Neck Surgery, Shanghai Ninth People's Hospital, Shanghai Jiao Tong University School of Medicine, Shanghai, China; [2]Ear Institute, Shanghai Jiao Tong University School of Medicine, Shanghai, China; [3]Shanghai Key Laboratory of Translational Medicine on Ear and Nose Diseases, Shanghai, China; [4]Department of Neuroscience, University of Virginia, Charlottesville, United States

**Abstract** The MRTF–SRF pathway has been extensively studied for its crucial role in driving the expression of a large number of genes involved in actin cytoskeleton of various cell types. However, the specific contribution of MRTF–SRF in hair cells remains unknown. In this study, we showed that hair cell-specific deletion of *Srf* or *Mrtfb*, but not *Mrtfa*, leads to similar defects in the development of stereocilia dimensions and the maintenance of cuticular plate integrity. We used fluorescence-activated cell sorting-based hair cell RNA-Seq analysis to investigate the mechanistic underpinnings of the changes observed in *Srf* and *Mrtfb* mutants, respectively. Interestingly, the transcriptome analysis revealed distinct profiles of genes regulated by *Srf* and *Mrtfb*, suggesting different transcriptional regulation mechanisms of actin cytoskeleton activities mediated by *Srf* and *Mrtfb*. Exogenous delivery of calponin 2 using Adeno-associated virus transduction in *Srf* mutants partially rescued the impairments of stereocilia dimensions and the F-actin intensity of cuticular plate, suggesting the involvement of *Cnn2*, as an *Srf* downstream target, in regulating the hair bundle morphology and cuticular plate actin cytoskeleton organization. Our study uncovers, for the first time, the unexpected differential transcriptional regulation of actin cytoskeleton mediated by *Srf* and *Mrtfb* in hair cells, and also demonstrates the critical role of SRF–CNN2 in modulating actin dynamics of the stereocilia and cuticular plate, providing new insights into the molecular mechanism underlying hair cell development and maintenance.

**\*For correspondence:**
tingtingdu_jei@163.com (TTD);
wuhao@shsmu.edu.cn (HW)

†These authors contributed equally to this work

**Competing interest:** The authors declare that no competing interests exist.

## Editor's evaluation

This important study provides convincing evidence implicating two transcription factors in the regulation of the actin cytoskeleton that shapes the mechanosensory hair bundles of the inner ear's hair cells. Although the mechanistic understanding of their operation remains incomplete, the work will be of interest to biologists interested in the development and maintenance of the hair bundle.

## Introduction

The actin cytoskeleton constitutes the structural basis of all features required for hair cell mechanoreceptor function, including the hair bundle and the cuticular plate (*Drummond et al., 2012*; *Furness et al., 2005*; *Tilney et al., 1989*). Actin-based stereocilia form rows of staircase-like structures with precise control of length and diameter on the apical surface of hair cells (*Tilney et al., 1992*;

*Kaltenbach et al., 1994*; *Krey et al., 2020*). Proper development of stereocilia dimensions determines the mechanical response of hair bundles to sound stimuli or head motions. The cuticular plate consists of a dense network of actin filaments crosslinked by actin-binding proteins (*DeRosier and Tilney, 1989*; *Mahendrasingam et al., 1998*; *Pollock and McDermott, 2015*), providing a mechanical foundation and rigidity for the stereocilia (*Furness et al., 2008*; *Self et al., 1998*). Disruption of the F-actin mesh integrity of cuticular plate results in hearing impairments (*Du et al., 2019*; *Liu et al., 2019*). Understanding how the hair cell governs its actin cytoskeleton is thus critical for understanding the development, function, and maintenance of the hair bundle and the cuticular plate. Substantial emerging evidence has revealed many genes indispensable for stereocilia elongation and widening, including actin crosslinkers, capping proteins, and myosins. Similarly, the complex cytoskeletal structure of cuticular plate is regulated by actins and actin crosslinkers. However, the general molecular mechanism of transcriptional regulation of actin cytoskeletal dynamics in stereocilia and cuticular plate remains poorly understood.

Serum response factor (SRF) is a highly conserved and ubiquitously expressed transcription factor that belongs to the MADS-box protein family. SRF activates the expression of target genes mainly through two disparate cofactors, TCFs (ternary complex factors) and MRTFs (myocardin-related transcription factors). These two families of cofactors compete for a common binding site of SRF, enabling *Srf* to direct the expression of different sets of target genes. *Srf* directly regulates immediate early genes mostly associated with cell proliferation, signaling, and transcription in coordination with TCFs (*Shaw et al., 1989*). Three MRTFs, including the cardiovascular system-specific expressed member Myocardin, and widely expressed MRTFA and MRTFB, are known to interact with SRF. The MRTF–SRF pathway has been extensively studied genetically in many systems, and it drives a large number of genes involved in actin cytoskeletal activities, contractility, and muscle differentiation (*Olson and Nordheim, 2010*; *Posern and Treisman, 2006*; *Wang et al., 2001*; *Wang et al., 2004*). Phenotypically, inactivation of *Srf* or *Mrtf*s often exhibits similar impairments. Conditional brain *Mrtfa/b* double-knockout mice show abnormalities in neuronal migration and neurite outgrowth that recapitulated the defects reported for brain-specific deletion of *Srf* (*Alberti et al., 2005*; *Knöll et al., 2006*). Podocyte-specific loss of both *Mrtf*s mimics the *Srf* deficiency phenotype in the maintenance of podocyte structure and function (*Guo et al., 2018*). During hematopoietic development, inactivation of either *Srf* or both *Mrtf*s in HSC progenitor cells leads to defects in their colonization of the bone marrow (*Costello et al., 2015*). However, there is also evidence showing that MRTFs can act as cofactor of transcriptional factors other than SRF and impact on the expression of genes involved in actin cytoskeletal activity, such as cell motility (*Kim et al., 2017*; *Liao et al., 2014*; *Xing et al., 2015*).

Some well-known proteins encoded by MRTF–SRF target genes related to actin cytoskeleton in other cell types are essential for hair cell actin-based cytoskeleton structure. For example, structure components of F-actin, ACTB, and ACTG1, are required for stereocilia maintenance (*Belyantseva et al., 2009*; *Perrin et al., 2013*; *Perrin et al., 2010*). F-actin severing protein CFL1 functions in stereocilia dimensions (*McGrath et al., 2021*), and actin-binding protein gelsolin is crucial for maintaining the cohesion of the hair bundle structure (*Olt et al., 2014*). However, it is unclear whether MRTF–SRF-mediated transcriptional regulation is required for hair cell actin-based cytoskeleton structure and function.

Our results demonstrate that in hair cells, both *Srf* and *Mrtfb*, but not *Mrtfa*, are necessary for the development and maintenance of actin-based cytoskeleton structures, such as the hair bundle and the cuticular plate. The specific deletion of *Srf* in hair cells resulted in a significant reduction in stereocilia length and diameter, as well as a decrease in F-actin intensity in the cuticular plate. In the absence of *Mrtfb*, we observed alterations in stereocilia dimensions and a loss of cuticular plate integrity, which are similar but less severe than those observed in *Srf* mutants. Furthermore, *Mrtfb* cKO mice developed early-onset hearing loss, in addition to the impairments in the hair bundle and cuticular plate. Differentially expressed genes (DEGs) analysis revealed distinct molecular consequences of *Srf* or *Mrtfb* deficiency in hair cell transcriptome, indicating that *Mtrfb* may function in an *Srf*-independent manner. Finally, we identified the actin-binding protein CNN2 as a downstream effector of *Srf* that is involved in regulating stereocilia dimensions and the organization of the F-actin network in the cuticular plate. These findings highlight the critical role of *Srf* and *Mrtfb* in the development and maintenance of hair cells, and shed light on the molecular mechanisms underlying actin cytoskeleton regulation in these cells.

## Results

### *Srf* cKO mice have defects in the cuticular plate

As previously described, deficiency of the transcription factor *Srf* (*Srf* $^{-/-}$) causes embryonic lethality due to severe gastrulation defects (*Arsenian et al., 1998*). To circumvent this limitation, we utilized mice expressing the recombinase Cre under the control of *Atoh1* promoter, permitting deletion of floxed *Srf* in hair cells. Offspring with the *Srf* $^{fl/fl}$; *Atoh1*$^{Cre}$ genotype (hereafter referred to as *Srf* cKO) was born at normal mendelian ratios, but displayed reduced body weight and a relatively weak appearance, and mostly died in the third postnatal week (*Figure 1—figure supplement 1A, B*), potentially due to *Atoh1-Cre* recombinase activity in the intestine and the brain (*Shroyer et al., 2007*; *Yang et al., 2010*; *Alberti et al., 2005*). However, gross morphology of the inner ear was not significantly affected (*Figure 1—figure supplement 1C*). Immunocytochemistry confirmed that SRF was detectable in all cell types of the organ of Corti at E16 (data not shown). To validate hair cell-specific knockout of *Srf*, we confirmed that SRF protein expression became undetectable at P0 in hair cells but not in supporting cells (*Figure 1—figure supplement 1D*). We further confirmed that *Srf* mRNA level in hair cells isolated from *Srf* cKO pups was significantly reduced compared to control hair cells (*Figure 1—figure supplement 1E*).

Upon closer inspection, we found that the cuticular plate of inner hair cells (IHCs) and outer hair cells (OHCs) from *Srf* cKO mice displayed a significant reduction in the amount of F-actin as indicated by phalloidin staining (*Figure 1A, B*). LMO7, an actin-binding protein important for F-actin network organization (*Du et al., 2019*), displayed a more punctate distribution in the cuticular plate (*Figure 1C*). We also examined the rootlets, which are composed of tightly packed actin filaments and form the stereocilia insertion sites into the cuticular plate. The rootlets are wrapped with TRIOBP, an actin-bundling protein that is an essential component of the rootlets (*Kitajiri et al., 2010*). Although the spatial organization of the rootlets appeared normal, the length of TRIOBP-positive rootlets was significantly reduced in *Srf* cKO mice, along with the intensity of TRIOBP staining (*Figure 1D, E*). Interestingly, the deficit in the integrity of the cuticular plate F-actin mesh accompanied by shortened rootlets also appeared in *Lmo7*-deficient mice as described in a previous study (*Du et al., 2019*). This similarity suggests that F-actin network in the cuticular plate is critical for the proper formation of the rootlets, although the causal connection between them remains unknown. To further investigate the F-actin meshwork abnormalities in the cuticular plate, we quantified the volume of the phalloidin-labeled cuticular plates after 3D reconstruction using Imaris software and also conducted an ultrastructural analysis using transmission electron microscopy (TEM). The results of the cuticular plate volume (CP volume) (*Figure 1F, G*) were consistent with the relative F-actin fluorescence intensity change of the cuticular plate (*Figure 1A, B*). The TEM results demonstrated that the cuticular plates in *Srf* cKO mice were markedly thinner compared to control in IHCs at P10 (*Figure 1H*). Although TEM images did not indicate an obvious reduction in the thickness of F-actin meshwork in P10 *Srf* cKO OHCs, the vesicle-like structures were often seen in the cuticular plate of the mutants (*Figure 1I*). The significant reduction of F-actin intensity indicating by phalloidin staining in *Srf* cKO OHCs (*Figure 1B*) may partially be caused by the vesicle-like structures occupation in the cuticular plate. The deformation of the overlying plasma membrane above the cuticular plate of OHCs were also revealed by scanning electron microscopy (SEM) in mutants (*Figure 2F*, as indicated by arrows).

### *Srf* is required for the development and maintenance of stereocilia dimensions

We conducted a systematic examination of hair bundle morphology using immunocytochemistry and SEM at different developmental stages to assess the effects of *Srf* deficiency. We used phalloidin staining to visualize actin core of stereocilia and measured the length of stereocilia. At P5, the staircase-like architecture of *Srf* cKO hair bundles developed well, but the length of row 1 (the longest row) IHC stereocilia was slightly reduced, which worsened by P10 (*Figure 2A, B* and *Figure 2—figure supplement 1A*). SEM further revealed that the length of row 1 IHC, the length of row 1 OHC, and the width of row 1 IHC stereocilia were significantly reduced in mutants at P10, but no statistical difference was found for the width of row 1 OHC stereocilia yet (*Figure 2C–E*). In rare cases, a few cochleae of *Srf* cKO mice survived past 2 weeks and were harvested at P15 and P24. Immunocytochemistry showed that the length and width of row 1 IHC stereocilia were obviously reduced in mutants at P15 and P24, as well as the width of stereocilia in utricles (*Figure 2A* and *Figure 2—figure supplement*

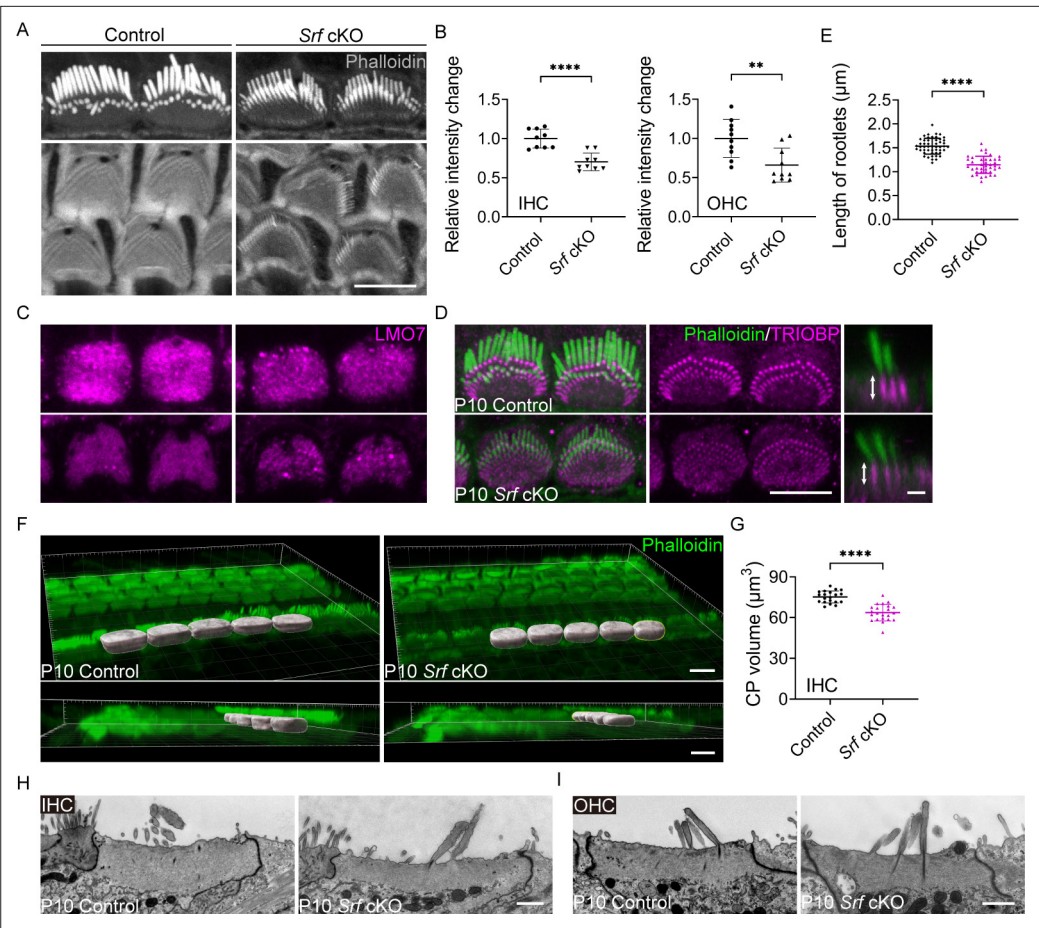

**Figure 1.** *Srf* cKO mice have defects in the cuticular plate. (**A**) Reduced phalloidin staining in the cuticular plates of *Srf* cKO hair cells (HCs) at P10. (**B**) Quantification of phalloidin reactivity in the cuticular plates of HCs in control and *Srf* cKO mice at P10. The relative intensity changes are the ratio of phalloidin intensity of mutant and control cuticular plates. Analyzed numbers (cells, animals): inner hair cells (IHCs), control (68, 6), *Srf* cKO (67, 6). Outer hair cells (OHCs), control (240, 6), *Srf* cKO (269, 6). (**C**) LMO7 immunostaining in HCs of control and *Srf* cKO at P10. (**D**) TRIOBP immunostaining in IHCs of control and *Srf* cKO at P10. Side views of TRIOBP labeling in IHCs are on the right. (**E**) Quantification of rootlet length of the longest row of IHC stereocilia (as shown by arrows in D) in control and *Srf* cKO at P10. Analyzed numbers (stereocilia, cells, animals): control (50, 14, 4), *Srf* cKO (41, 12, 4). (**F**) Imaris 3D reconstruction of phalloidin-labeled cuticular plates in control and *Srf* cKO mice at P10. The contour of the cuticular plate outlined by a yellow line in *Srf* cKO represents the area used to calculate the volume of the cuticular plate. (**G**) Quantification of cuticular plate volume of control and *Srf* cKO IHCs. Analyzed numbers (cells, animals): control (20, 5), *Srf* cKO (22, 5). (**H, I**) Transmission electron microscopy (TEM) analysis of cuticular plates of IHCs and OHCs in control and *Srf* cKO at P10. In D, scale bar for the side-view represents 1 μm. In H and I, the scale bars represent 1 μm. Scale bars in other panels represent 5 μm. Error bars indicate standard deviation (SD), p values were derived from two-tailed unpaired Student's *t*-test, ****p-value <0.0001, **p-value <0.01.

The online version of this article includes the following figure supplement(s) for figure 1:

**Figure supplement 1.** Conditional knockout of *Srf* in mice caused growth defects.

---

*1B*). Statistical analysis of SEM images confirmed that the width of rows 1 and 2 IHC stereocilia was drastically reduced by 40% in *Srf* cKO at P15 (*Figure 2G, H*).

The loss of *Srf* also affected the regression of microvilli at the medial side of the hair bundle. During early hair bundle development, a subset of microvilli on the apical surface of the hair cell differentiate into stereocilia while the rest regress (*Tilney et al., 1992*). At P15, numerous microvilli still persisted in *Srf* cKO IHCs, indicating a lack of mature hair bundle characteristics, but this was not the case for *Srf* cKO OHCs (*Figure 2F, G*). The kinocilium, a microtubule-based cilium that is essential for the polarity of the developing hair bundle, degenerates in mouse cochlear hair cells from base to apex

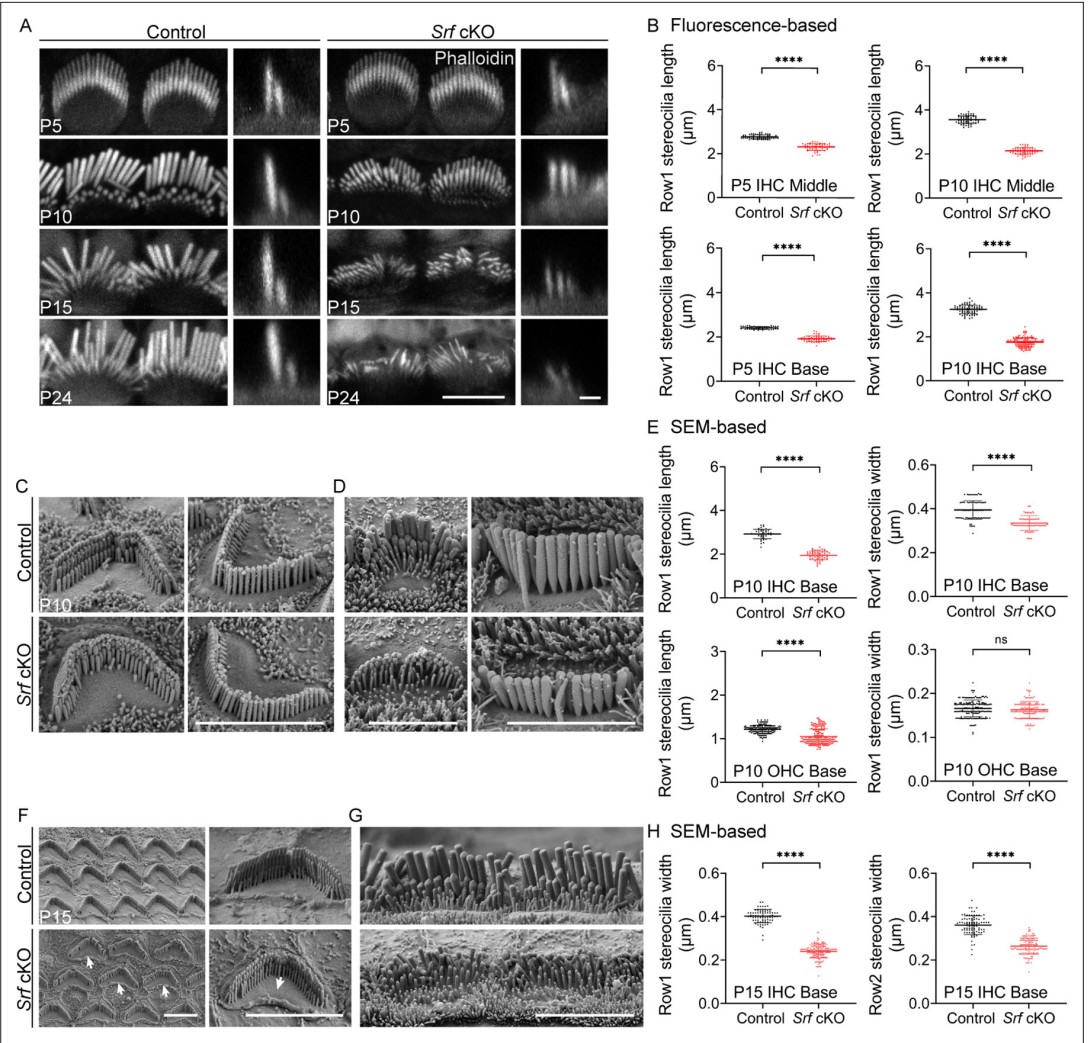

**Figure 2.** The defects of stereocilia dimensions in *Srf* cKO mice. (**A**) Phalloidin-stained inner hair cell (IHC) bundles of control and *Srf* cKO at different ages. En face views (scale bar, 5 µm) are on the left, and side views (scale bar, 1 µm) are on the right. (**B**) Fluorescence-based quantification of the length of row 1 IHC stereocilia at different cochlear positions, at P5 and P10. Analyzed numbers (stereocilia, cells, animals): middle IHCs, P5 control (60, 30, 5), P5 *Srf* cKO (62, 30, 5), P10 control (61, 25, 6), P10 *Srf* cKO (62, 26, 6). Basal IHCs, P5 control (41, 21, 5), P5 *Srf* cKO (53, 24, 5), P10 control (74, 28, 6), P10 *Srf* cKO (153, 53, 6). (**C, D**) Scanning electron microscopy (SEM) of outer hair cells (OHCs) and IHCs in control and *Srf* cKO mice at P10. (**E**) SEM-based quantification of row 1 stereocilia length and width in basal HCs at P10. Analyzed numbers: lengths: IHCs, control (42, 20, 5), *Srf* cKO (52, 18, 5). OHCs, control (149, 43, 5), *Srf* cKO (165, 42, 5). Widths: IHCs, control (88, 22, 5), *Srf* cKO (81, 20, 5). OHCs, control (164, 45, 5), *Srf* cKO (147, 42, 5). (**F, G**) SEM of OHCs and IHCs in control and *Srf* cKO mice at P15. (**H**) SEM-based quantification of rows 1 and 2 stereocilia width in basal IHCs at P15. Analyzed numbers: row 1, control (76, 11, 3), *Srf* cKO (150, 15, 3). Row 2, control (85, 11, 3), *Srf* cKO (134, 14, 3). Scale bars in C, D, F and G represent 5 µm. Error bars indicate standard deviation (SD), p values were derived from two-tailed unpaired Student's *t*-test, ****p value<0.0001.

The online version of this article includes the following figure supplement(s) for figure 2:

**Figure supplement 1.** *Srf* cKO mice have defects in hair bundles.

at P8 and disappears completely around P12 (**Haag et al., 2018**; **Leibovici et al., 2005**). Confocal microscopy images showed that more kinocilia remained in *Srf* cKO IHCs compared to the controls at P10 (*Figure 2—figure supplement 1C, D*). Presumably, this also indicates that *Srf* cKO hair cells were developmentally less mature, although the physiological significance of kinocilia degeneration is still unclear. We conclude that SRF is required for the development as well as the maintenance of the F-actin-enriched hair bundle and the cuticular plate.

## Altered expression pattern of proteins required for stereocilia dimensions in *Srf* cKO

Given that the stereocilia dimensions were affected in *Srf* cKO hair bundle, we quantified the localization of several key proteins known to be critical for hair bundle development. ESPN1, an isoform of ESPN, which is an actin-binding and bundling protein, is required for the elongation of hair cell stereocilia (*Ebrahim et al., 2016*; *Rzadzinska et al., 2005*; *Salles et al., 2009*). Similarly, EPS8 and GNAI3, which are components of the row 1 complex, are essential for controlling the elongation of the tallest row of stereocilia during hair bundle morphological development (*Tarchini et al., 2016*; *Zampini et al., 2011*). Using immunofluorescence, we quantified the tip signal of these proteins in row 1 stereocilia at P4 and P10. The dramatic decreases of the relative row 1 tip fluorescence signal intensity for ESPN1 and EPS8, but not GNAI3, started since P4 in *Srf* cKO IHCs. The decrease of GNAI3 in *Srf* cKO IHCs was observed soon after at P10 (*Figure 3A–F*). In addition to the reduced row 1 tip localization of ESPN1, EPS8, and GNAI3, the irregular labeling in different stereocilia of the same row were also observed in these tested proteins in *Srf* cKO mice (*Figure 3G–I*). Meanwhile, the row 2 tip signals of ESPN1, EPS8, and GNAI3 were partially maintained in *Srf* cKO IHCs compared to controls, although the expression of EPS8 and GNAI3 were abundant in row 1 tip and absent in row 2 tip in controls at P10 (*Figure 3J–L*). These data suggested that, initially, all the tested proteins responsible for stereocilia elongation were not efficiently targeting to row 1 stereocilia tips in *Srf* cKO mice. During stereocilia development, the distribution of EPS8 and GNAI3 between rows 1 and 2 stereocilia were also affected. We also quantified the relative fluorescence signals of FSCN2, an abundantly expressed actin-crosslinking protein in stereocilia (*Perrin et al., 2013*), within the stereocilia shaft. The fluorescence levels were significantly reduced by 20% in *Srf* cKO IHCs at P10 (*Figure 3M, N*). In summary, *Srf* deficiency in hair cells compromises the integrity of F-actin mesh in the cuticular plate and reduces the dimensions of stereocilia.

## *Mrtfb*, but not *Mrtfa* is essential for the integrity of the cuticular plate and the development of stereocilia dimensions

Previous studies have identified *Mrtfa* and *Mrtfb*, the members of MRTF family, as having low levels of transcripts during inner ear development (*Cai et al., 2015*; *Elkon et al., 2015*; *Scheffer et al., 2015*). In the present study, we employed RNAscope staining to confirm the ubiquitous expression of *Mrtfa* and *Mrtfb* in E16 cochlea, including in hair cells (*Figure 4—figure supplement 1A* and *Figure 4—figure supplement 2A*). Based on the abnormality of hair cell actin cytoskeleton in *Srf* mutants, we aimed to investigate if *Mrtfa* or *Mrtfb* acts as a cofactor of *Srf* to regulate actin cytoskeleton genes involved in the development of cuticular plate and stereocilia. To achieve hair cell-specific deletion of *Mrtfa* or *Mrtfb*, we generated mice carrying floxed alleles of *Mrtfa* and *Mrtfb*, respectively (*Figure 4—figure supplement 1B* and *Figure 4—figure supplement 2B*), using CRISPR/Cas9 genome editing. Crossing these mice to *Atoh1-Cre* line resulted in the production of *Mrtfa* cKO and *Mrtfb* cKO mice, which were viable and fertile with no overt phenotype. *Mrtfa* mRNA and protein levels, as well as MRTFB protein immunoreactivity, were partially reduced in hair cells of respective knockout mice (*Figure 4—figure supplement 1C, D* and *Figure 4—figure supplement 2C, D*). We measured the intensity of F-actin in the cuticular plate and the length of row 1 IHC stereocilia in *Mrtfa* mutants and observed no significant changes compared to controls at P26 (*Figure 4—figure supplement 1E–G*). Additionally, auditory brainstem responses (ABRs) measurements and distortion product otoacoustic emissions (DPOAEs) recordings indicated normal hearing of *Mrtfa* cKO mice at P26 (*Figure 4—figure supplement 1H*).

In contrast to the *Mrtfa* cKO mice, *Mrtfb* cKO mice exhibited significant changes in the integrity of the cuticular plate and the dimensions of stereocilia. Specifically, a notable decrease in F-actin intensity was observed in the cuticular plate of *Mrtfb* cKO OHCs, with a trend toward reduced F-actin intensity noted in the cuticular plate of *Mrtfb* cKO IHCs (*Figure 4A, B*). Additionally, the length of TRIOBP-positive rootlets was reduced in the mutants, despite no observed changes in TRIOBP staining intensity (*Figure 4C, D*). The results of the CP volume in the control and *Mrtfb* cKO mice (*Figure 4E, F*) were consistent with the relative F-actin fluorescence intensity change of the cuticular plate (*Figure 4A, B*). Using TEM analysis, the results demonstrated that the cuticular plates in *Mrtfb* cKO mice were markedly thinner compared to control in OHCs at P10. The vesicle-like structures occupation in the cuticular plate were also observed in the mutants. The reduction of the cuticular plate thickness in

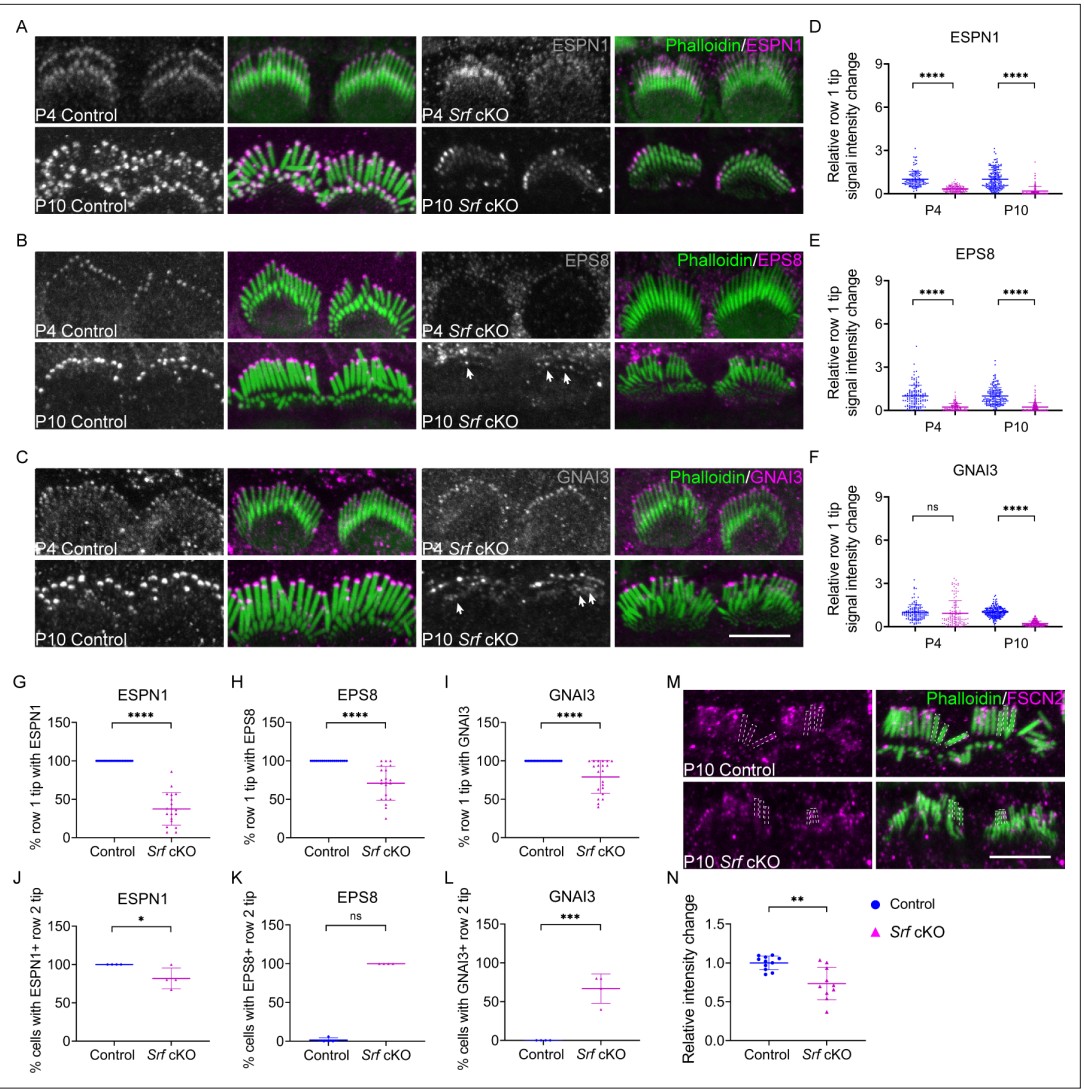

**Figure 3.** Altered distribution of tip proteins and actin crosslinker in *Srf* cKO mice. (**A–C**) Immunofluorescence localization of ESPN1, EPS8, and GNAI3 in inner hair cells (IHCs) of control and *Srf* cKO at P4 and P10. Arrows indicate proteins staining on row 2 tips. Left panels are proteins staining separately from the phalloidin (in grayscale). (**D–F**) Quantitation of immunoreactivity of ESPN1, EPS8, and GNAI3 on IHCs stereocilia tips of control and *Srf* cKO, at P4 and P10. Analyzed numbers (stereocilia, cells, animals): ESPN1, P4 control (120, 12, 4), P4 *Srf* cKO (120, 12, 4), P10 control (180, 18, 5), P10 *Srf* cKO (170, 17, 5). EPS8, P4 control (120, 12, 5), P4 *Srf* cKO (110, 11, 5), P10 control (170, 17, 5), P10 *Srf* cKO (200, 20, 5). GNAI3, P4 control (120, 12, 4), P4 *Srf* cKO (110, 12, 4), P10 control (190, 19, 5), P10 *Srf* cKO (180, 18, 5). (**G–I**) Percentage of row 1 tip with ESPN1, EPS8, or GNAI3 staining to all row 1 stereocilia in IHCs of control and *Srf* cKO at P10. (**J–L**) Percentage of IHCs with ESPN1, EPS8, or GNAI3 staining on row 2 tips of control and *Srf* cKO at P10. Analyzed numbers in G–L (cells, animals): ESPN1, control (23, 5), *Srf* cKO (19, 5). EPS8, control (19, 5), *Srf* cKO (21, 5). GNAI3, control (25, 5), *Srf* cKO (21, 5). (**M**) FSCN2 immunostaining in control and *Srf* cKO IHCs. (**N**) Quantification of FSCN2 reactivity in IHC stereocilium shafts at P10. Analyzed numbers: control (110, 11, 5), *Srf* cKO (100, 10, 5). Scale bars, 5 µm. Error bars indicate standard deviation (SD), p values were derived from two-tailed unpaired Student's *t*-test, ****p-value <0.0001, ***p-value <0.001, **p-value <0.01, and *p-value <0.05.

---

*Mrtfb* cKO mice was more pronounced at P26. For IHCs, a decrease of the cuticular plate thickness in *Mrtfb* cKO was observed until P26 (*Figure 4G, H*). Immunocytochemistry was then used to examine hair bundle morphology at various developmental stages, from early postnatal to adult mice. At P5, no difference in row 1 IHC stereocilia length was observed between the control and *Mrtfb* cKO mice. However, the length of row 1 IHC stereocilia was significantly reduced at P10 and P26 (*Figure 5A, B*). While the staircase-like architecture of *Mrtfb* cKO OHC hair bundles formed adequately at P10,

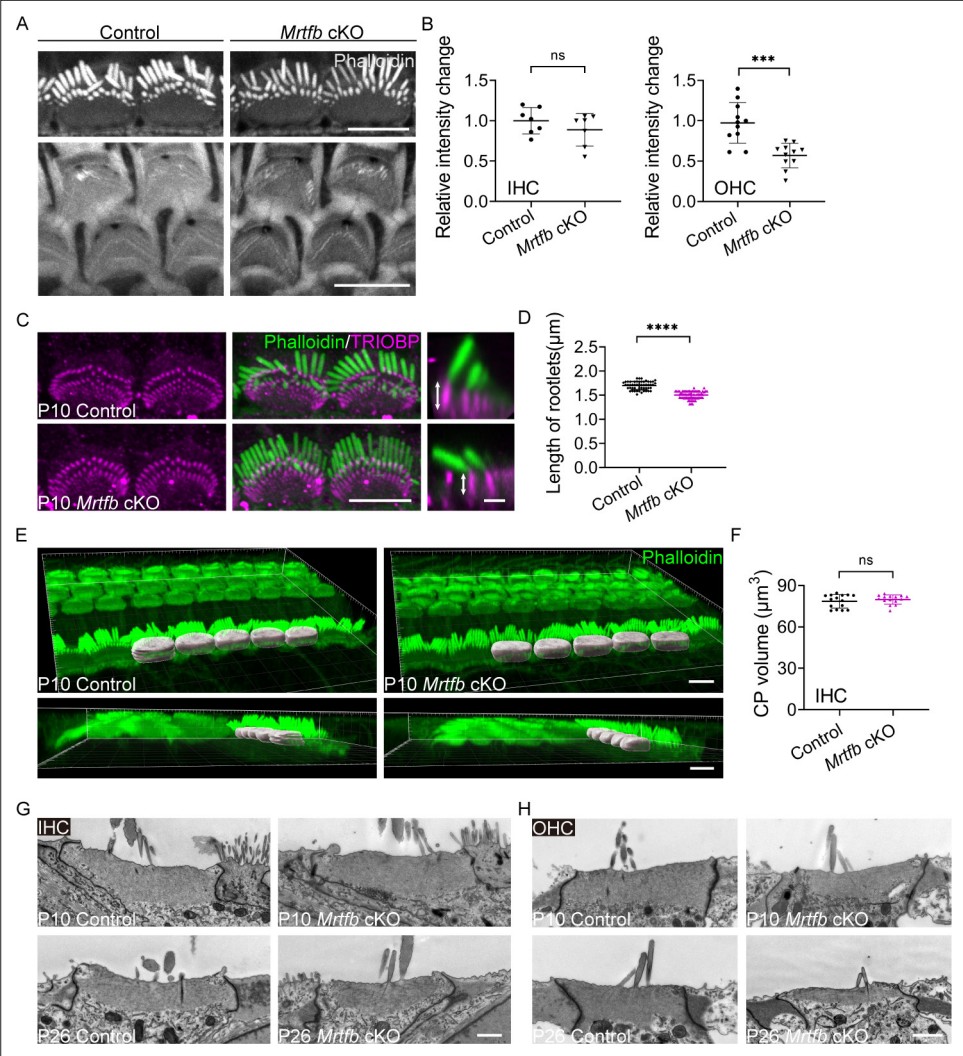

**Figure 4.** *Mrtfb* cKO mice have defects in the cuticular plates. (**A**) Reduced phalloidin staining in the cuticular plates of *Mrtfb* cKO HCs at P10. (**B**) Quantification of phalloidin reactivity in the cuticular plates of inner hair cells (IHCs) and outer hair cells (OHCs) at P10. Analyzed numbers (cells, animals): IHCs, control (60, 6), *Mrtfb* cKO (63, 6). OHCs, control (261, 6), *Mrtfb* cKO (265, 6). (**C**) TRIOBP immunostaining of IHCs at P10. Side views of TRIOBP labeling in IHCs are on the right. (**D**) Quantification of length of row 1 IHC stereocilia rootlets (as shown by arrows in C). Analyzed numbers (stereocilia, cells, animals): control (52, 16, 4), *Mrtfb* cKO (63, 23, 4). (**E**) Imaris 3D reconstruction of phalloidin-labeled cuticular plates in control and *Mrtfb* cKO mice at P10. (**F**) Quantification of cuticular plate volume of control and *Mrtfb* cKO IHCs. Analyzed numbers (cells, animals): control (15, 4), *Mrtfb* cKO (15, 4). (**G, H**) Transmission electron microscopy (TEM) analysis of cuticular plates of IHCs and OHCs in control and *Mrtfb* cKO at P10 and P26. In C, scale bar for the side-view represents 1 µm. In G and H, the scale bars represent 1 µm. Scale bars in other panels represent 5 µm. Error bars indicate standard deviation (SD), p values were derived from two-tailed unpaired Student's *t*-test, ****p value<0.0001, and ***p value<0.001.

The online version of this article includes the following figure supplement(s) for figure 4:

**Figure supplement 1.** No obvious defects in HCs and hearing function of *Mrtfa* cKO mice.

**Figure supplement 2.** Conditional knockout of *Mrtfb* in mice.

---

they progressively degenerated throughout development (*Figure 5—figure supplement 1A*). This reduction in stereocilia length and width was further confirmed through scanning electron microscope analysis of stereocilia morphologies at P10 and P42 (*Figure 5C–H*). Specifically, a slight reduction in stereocilia width was observed in OHCs, with the length of stereocilia remaining intact in mutants at P10 (*Figure 5C, E*). However, by P42, the stereocilia of OHCs had severely degenerated (*Figure 5F*). For IHCs, both the length and width of row 1 stereocilia showed significant decreases in *Mrtfb* mutants

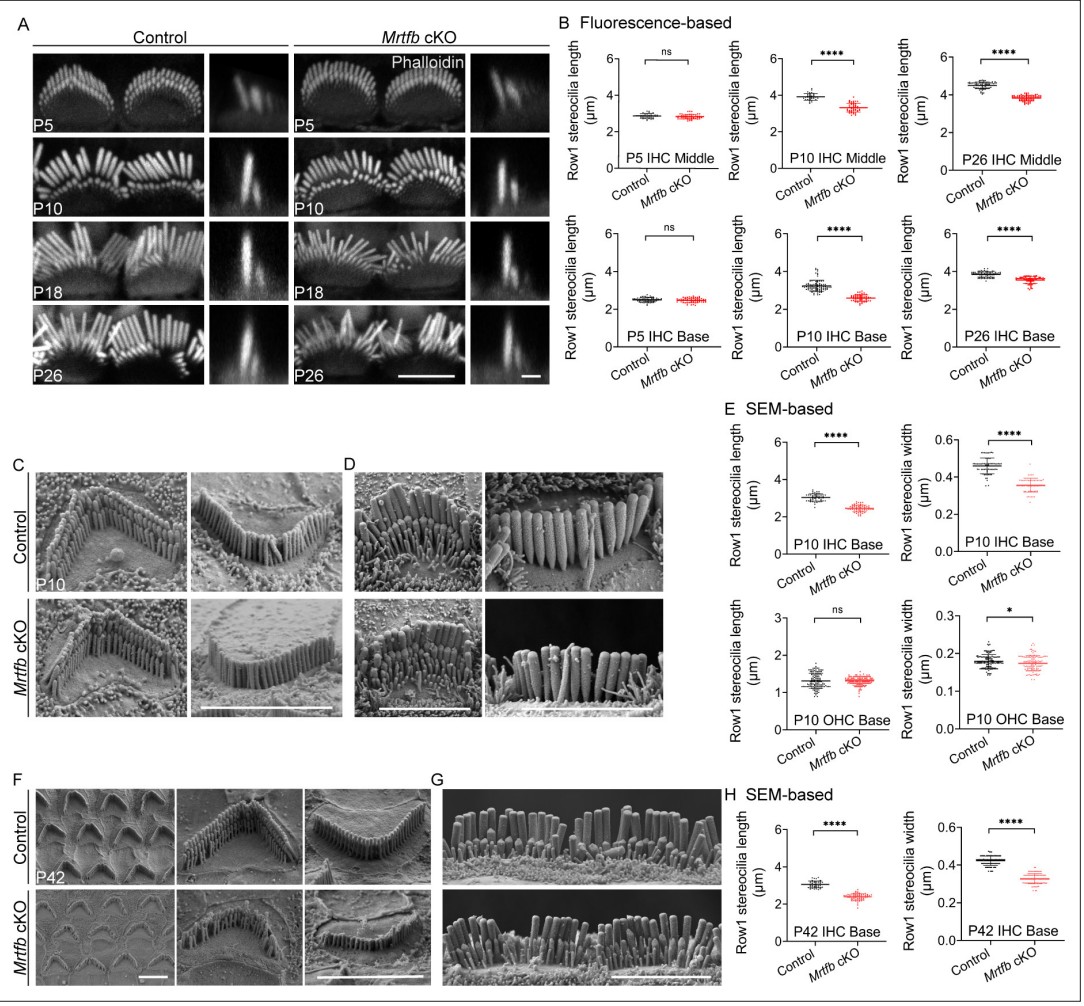

**Figure 5.** The defects of stereocilia dimensions in *Mrtfb* cKO mice. (**A**) Phalloidin staining of inner hair cell (IHC) bundles at different ages. En face views (scale bar, 5 µm) are on the left, and side views (scale bar, 1 µm) are on the right. (**B**) Fluorescence-based quantification of row 1 IHC stereocilia length at different cochlear positions and ages. Analyzed numbers: middle IHCs, P5 control (41, 20, 6), P5 *Mrtfb* cKO (53, 23, 6), P10 control (80, 35, 6), P10 *Mrtfb* cKO (36, 18, 6), P26 control (73, 23, 5), P26 *Mrtfb* cKO (71, 24, 5). Basal IHCs, P5 control (65, 28, 6), P5 *Mrtfb* cKO (60, 25, 6), P10 control (83, 39, 6), P10 *Mrtfb* cKO (48, 22, 6), P26 control (74, 25, 5), P26 *Mrtfb* cKO (73, 24, 5). (**C, D**) Scanning electron microscopy (SEM) of outer hair cells (OHCs) and IHCs at P10. (**E**) SEM-based quantification of row 1 stereocilia length and width in basal HCs of control and *Mrtfb* cKO at P10. Analyzed numbers: lengths: IHCs, control (50, 20, 6), *Mrtfb* cKO (59, 20, 6). OHCs, control (127, 33, 6), *Mrtfb* cKO (132, 39, 6). Widths: IHCs, control (65, 22, 5), *Mrtfb* cKO (58, 22, 5). OHCs, control (135, 35, 5), *Mrtfb* cKO (118, 39, 5). (**F, G**) SEM of OHCs and IHCs at P42. (**H**) SEM-based quantification of row 1 stereocilia length and width in basal IHCs of control and *Mrtfb* cKO at P42. Analyzed numbers: lengths, control (42, 22, 6), *Mrtfb* cKO (63, 24, 6). Widths, control (86, 22, 6), *Mrtfb* cKO (92, 24, 6). Scale bars in C, D, F and G represent 5 µm. Error bars indicate standard deviation (SD), p values were derived from two-tailed unpaired Student's *t*-test, ****p value<0.0001, and *p value<0.05.

The online version of this article includes the following figure supplement(s) for figure 5:

**Figure supplement 1.** *Mrtfb* cKO mice have stereocilia impairments.

at P10 (*Figure 5D, E*) and P42 (*Figure 5G, H*). Notably, defects in the dimensions of stereocilia of utricle hair cells were also observed in *Mrtfb* cKO mice (*Figure 5—figure supplement 1B*).

## Altered expression pattern of proteins required for stereocilia dimensions in *Mrtfb* cKO

Given that the localization and expression pattern of ESPN1, EPS8, and GNAI3 were affected in *Srf* cKO hair bundle, we further investigated the expression pattern of these proteins in *Mrtfb* cKO at

P4, P10, and P15 during stereocilia development. As shown in *Figure 6*, an obvious decrease in the relative row 1 tip fluorescence signal intensity for ESPN1 was initially observed at P10 in the mutants. Compared to ESPN1, the obvious reduction of relative row 1 tip fluorescence signal intensity of EPS8 and GNAI3 appeared much later in *Mrtfb* cKO, around P15 (*Figure 6A–F*). The irregular labeling of ESPN1 and GNAI3 in different stereocilia of the same row were occasionally observed in *Mrtfb* cKO mice (*Figure 6G–I*). The distribution of these proteins between rows 1 and 2 stereocilia were not affected in *Mrtfb* cKO mice at P10 (*Figure 6J–L*). Additionally, The fluorescence signals of FSCN2 within the stereocilia shaft were dramatically reduced by 40% in *Mrtfb* cKO IHCs (*Figure 6M, N*).

By comparing the expression level, the row-specific distribution and the irregular labeling of the tested proteins in *Srf* cKO and *Mrtfb* cKO hair cells, the results suggested that during stereocilia development, the effects of SRF and MRTFB on tip proteins' localization patterns were fundamentally different, which were highly consistent with the severity of the morphological defects of stereocilia dimensions in *Srf* cKO and *Mrtfb* cKO hair cells.

### *Mrtfb* deficiency causes early-onset hearing loss

The hearing function in mice lacking *Mrtfb* in hair cells was evaluated using ABR measurements. At P26, the ABR thresholds of *Mrtfb* cKO mice were significantly elevated, with more pronounced effects at mid- to high frequencies ranging from 11 to 32 kHz. By P40, *Mrtfb* cKO mice developed profound hearing loss at all frequencies tested, with ABR thresholds that were 40–60 dB higher than those of the control group (*Figure 7A*). To assess OHC functions, we further recorded DPOAEs. Consistent with ABRs measurements, DPOAE thresholds of *Mrtfb* cKO mice were markedly elevated over the 11–32 kHz frequency range. From P40 onward, the DPOAE thresholds increased progressively for all frequencies (*Figure 7B*). The amplitude and latency of ABR wave 1 at 11.3 kHz were also analyzed. In comparison to normal hearing control mice, the amplitude of ABR wave 1 was reduced at 80–90 dB, and the latency of ABR wave 1 was significantly prolonged (*Figure 7C*).

In addition to the impairments of cuticular plate integrity and stereocilia dimensions, we sought to investigate the other potential factors contributing to the early-onset hearing loss observed in *Mrtfb*-deficient mice. Previous studies have shown that the actin cytoskeleton beneath the lateral plasma membrane of OHCs is necessary for maintaining membrane rigidity and regulating OHC electromotility (*Brownell et al., 2001*; *Holley and Ashmore, 1990*). Therefore, we examined whether *Mrtfb* is crucial for the development of OHC electromotility. Initially, we evaluated the expression level and localization of Prestin the motor protein essential for OHC electromotility, in *Mrtfb* mutants. Our results showed that the expression level and localization of Prestin were normal in *Mrtfb* mutants (*Figure 7D*). We then performed whole-cell patch-clamp recordings at P18 to evaluate the functional changes of OHCs in the absence of *Mrtfb*. We quantified the nonlinear capacitance, total number of Prestin ($Q_{max}$), cell sizes ($C_{lin}$), charge density ($Q_{sp}$), and voltage sensitivity ($V_h$), and found no significant differences between *Mrtfb* cKO and control mice (*Figure 7E–I*). These findings suggest that *Mrtfb* is not required for the development of OHC electromotility during early postnatal stages.

A dense network of F-actin has been observed to surround the ribbon synapse of IHCs using phalloidin staining (*Guillet et al., 2016*; *Vincent et al., 2015*). We investigatedthe effect of *Mrtfb* on the cochlear ribbon synapse by analyzing the expression of CtBP2, presynaptic $Ca_v1.3$ and postsynaptic GluR2. The number of CtBP2-positive synapses in *Mrtfb* mutants was comparable to controls, and normal co-localization of CtBP2 and $Ca_v1.3$, as well as CtBP2 and GluR2, was observed (*Figure 7J–L*). The F-actin mesh network at the ribbon synapse is known to play a crucial role in the spatial localization of $Ca_v1.3$ $Ca^{2+}$ channels and vesicle exocytosis in IHCs. Further investigation is necessary to examine the distance between CtBP2 and $Ca_v1.3$ and evaluate exocytosis in IHCs in the absence of *Mrtfb*.

To determine whether the hearing loss in *Mrtfb* cKO mice was caused by hair cell loss, we counted hair cell numbers using MYO7A immunoreactivity at P26 and P42. Our results indicated that hair cell numbers in *Mrtfb* cKO mice were nearly intact, except for a reduction in OHC numbers at the base of *Mrtfb* cKO cochleae at P42 (*Figure 7—figure supplement 1A, B*). This suggests that the early-onset hearing loss observed in *Mrtfb* mutants is not primarily caused by hair cell loss.

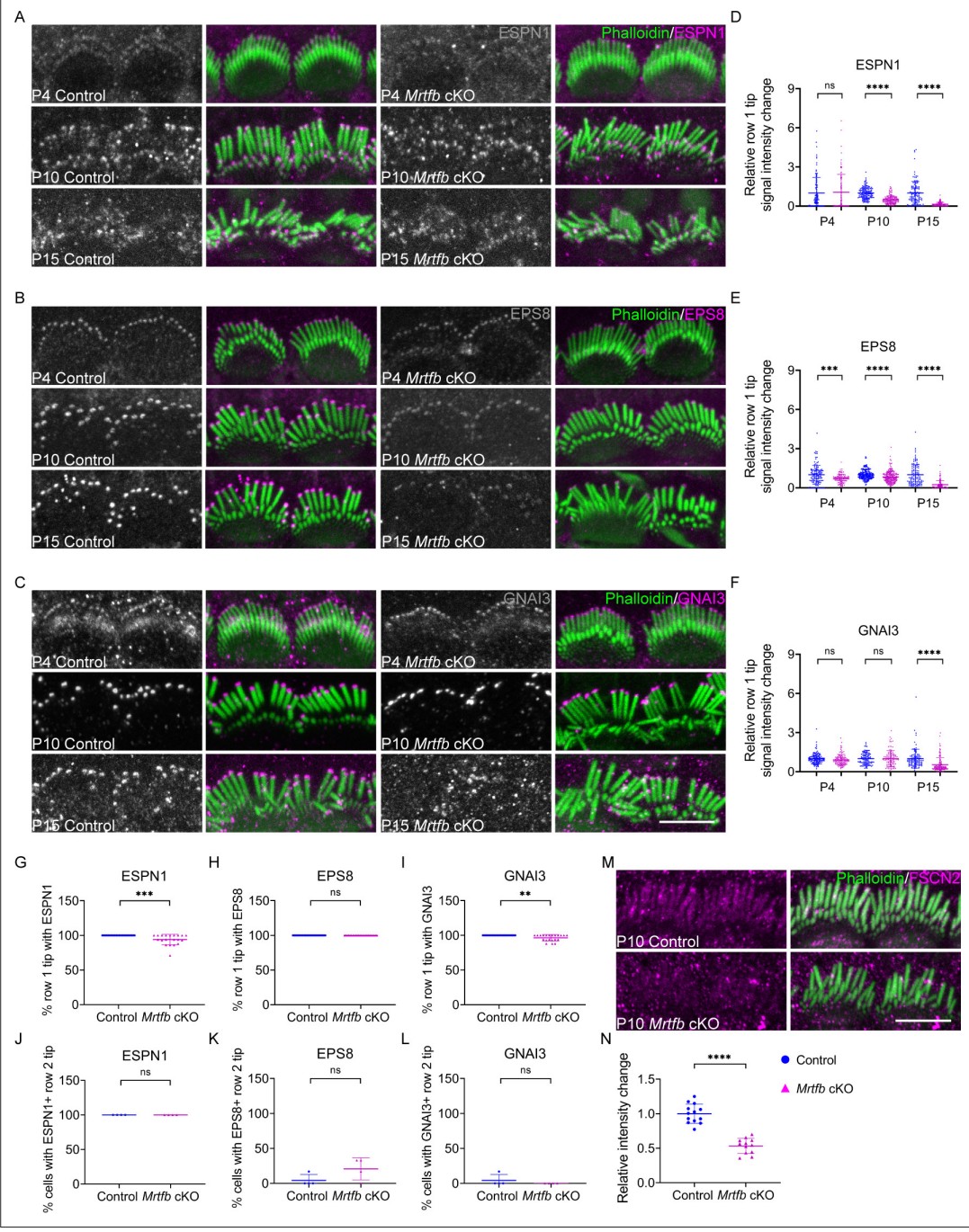

**Figure 6.** Altered distribution of tip proteins and actin crosslinker in *Mrtfb* cKO mice. (**A–C**) Immunofluorescence localization of ESPN1, EPS8, and GNAI3 in inner hair cells (IHCs) of control and *Mrtfb* cKO mice at different ages. Left panels are proteins staining separately from the phalloidin (in grayscale). (**D–F**) Quantitation of immunoreactivity of ESPN1, EPS8, and GNAI3 in IHCs of control and *Mrtfb* cKO at different ages. Analyzed numbers (stereocilia, cells, animals): ESPN1, P4 control (110, 11, 4), P4 *Mrtfb* cKO (110, 11, 4), P10 control (130, 13, 5), P10 *Mrtfb* cKO (140, 14, 5), P15 control (110, 11, 4), P15 *Mrtfb* cKO (120, 12, 4). EPS8, P4 control (120, 12, 5), P4 *Mrtfb* cKO (100, 10, 5), P10 control (190, 19, 5), P10 *Mrtfb* cKO (200, 20, 5), P15 control (110, 11, 5), P15 *Mrtfb* cKO (110, 11, 5). GNAI3, P4 control (120, 12, 5), P4 *Mrtfb* cKO (110, 11, 5) P10 control (100, 10, 4), P10 *Mrtfb* cKO (100, 10, 4), P15 control (100, 11, 4), P15 *Mrtfb* cKO (120, 11, 4). (**G–I**) Percentage of row 1 tip with ESPN1, EPS8, or GNAI3 staining to all row 1 stereocilia in IHCs of control and *Mrtfb* cKO at P10. (**J–L**) Percentage of IHCs with ESPN1, EPS8, or GNAI3 staining on the row 2 tips of control and *Mrtfb* cKO at P10. Analyzed numbers in G–L (cells, animals): ESPN1, control (24, 5), *Mrtfb* cKO (20, 5). EPS8, control (24, 5), *Mrtfb* cKO (24, 5). GNAI3, control

*Figure 6 continued on next page*

*Figure 6 continued*

(24, 4), *Mrtfb* cKO (21, 4). (**M**) FSCN2 immunostaining in control and *Mrtfb* cKO IHCs. (**N**) Quantification of FSCN2 reactivity in IHC stereocilium shafts at P10. Analyzed numbers (stereocilia, cells, animals): control (130, 13, 6), *Mrtfb* cKO (120, 12, 6). Scale bars, 5 µm. Error bars indicate standard deviation (SD), p values were derived from two-tailed unpaired Student's *t*-test, ****p value<0.0001, ***p value<0.001, and **p value<0.01.

## RNA-sequencing reveals the distinct profiles of genes regulated by *Srf* and *Mrtfb* in hair cells

Given that *Mrtfb* mutants exhibit a mild phenotype similar to the loss of *Srf* in actin-enriched structures such as the cuticular plate and hair bundles, we hypothesized that MRTFB–SRF axis regulates the actin-based cytoskeleton structures and function of hair cells. To investigate the molecular mechanism underlying the observed changes in hair bundle and cuticular plates in *Srf* and *Mrtfb* mutants, we performed RNA-sequencing of control and mutant hair cells. We utilized mice harboring *Atoh1-Cre*, *Rosa26* with *LSL-tdTomato* reporter (also known as *Ai9*) and floxed *Srf* or *Mrtfb* alleles. At P2, cochleae were dissected and dissociated into single cells, and tdTomato-positive hair cells were isolated by fluorescence-activated cell sorting (FACS) based on tdTomato fluorescence intensity (*Figure 8A*). We performed differential gene expression analysis ($p_{adj} < 0.05$ and $|\log_2(\text{fold change})| > 1$) (*Figure 8B*), and identified 110 up-regulated and 79 down-regulated DEGs in *Srf* cKO hair cells (the down-regulated genes hereinafter referred to as *Srf* targets), and 15 up-regulated and 105 down-regulated DEGs in *Mrtfb* cKO hair cells (the down-regulated genes hereinafter referred to as *Mrtfb* targets). To validate the RNA-Seq results, we selected ten *Srf* targets (*Srf*, *Actb*, *Actg1*, *Cfl1*, *Actr3*, *Myo1e*, *Thbs2*, *Ush2a*, *Camp*, and *Htra4*), eight *Mrtfb* targets (*Srf*, *Actb*, *Actg1*, *Cfl1*, *Actr3*, *Kif14*, *Rsf1*, and *Alkbh8*), and one unchanged gene *Myo7a* for RT-qPCR confirmation (*Figure 8C, D*). Using immunocytochemistry and RNAscope, we confirmed the reduction of CTGF (a *Srf* target), Pcdhb15 and Ctnna3 (*Mrtfb* targets) in hair cells consistent with RNA-Seq analysis results (*Figure 8F–H*). Interestingly, none of the down-regulated genes in *Srf* targets and *Mrtfb* targets were common, highly suggesting that in hair cells, *Srf* or *Mrtfb* may regulate the distinct profiles of genes that affect the development and function of the actin-based structures. To further confirm this hypothesis, we first quantified the relative expression level of known MRTF–SRF target genes (*Actb*, *Actg1*, *Cfl1*, and *Actr3*) that are essential for hair cell actin cytoskeleton (*Esnault et al., 2014*). Only *Actb* was reduced in *Srf* cKO hair cells, consistent with the immunoreactivity of ACTB (*Figure 8C–E*). We then summarized the relative expression level of all the reported target genes of MRTF–SRF axis with $p_{adj} < 0.05$ in our hair cell dataset (*Figure 8—figure supplement 1A*). The results showed that no genes were tightly coregulated by both *Srf* and *Mrtfb* in the context of hair cells.

We next performed a Gene Ontology (GO) enrichment analysis of down-regulated DEGs related to biological processes to better understand the role of *Srf* and *Mrtfb* in hair cells. In *Srf* mutants, the GO analysis revealed significant enrichment in five biological processes: regulation of cell growth, cell growth, regulation of growth, defense response, and growth. The DEGs enriched in the growth-associated GO terms included *Ctgf* and *Htra4*, while *Camp* and *Ngp* were enriched in the defense response term (*Figure 8I*). On the other hand, in *Mrtfb* mutants, the GO analysis identified a significant enrichment of genes associated with adhesion-related terms, such as cell adhesion, biological adhesion, homophilic cell adhesion via plasma membrane adhesion molecules, cell–cell adhesion via plasma membrane adhesion molecules, and cell–cell adhesion. The DEGs enriched in these GO terms included *Cdh19*, *Pcdhb7*, *Pcdhb12*, *Pcdhb15*, and *Ctnna3* (*Figure 8I*).

Although no significant enriched GO terms were found for cytoskeleton-related genes, a number of down-regulated DEGs were identified. We conducted a functional clustering analysis of down-regulated DEGs based on GO annotations to comprehensively summarize the cytoskeleton-related genes in *Srf* and *Mrtfb* targets. In *Srf* cKO, seven down-regulated DEGs were involved in cytoskeleton organization, including actin cytoskeleton organization, actin cytoskeleton reorganization, actin filament organization, and so on. For *Mrtfb* cKO, *Ctnna3* was the only gene involved in cytoskeleton organization and its ancestor term, actin filament organization. Moreover, in terms of molecular function, five down-regulated DEGs in *Srf* cKO and seven down-regulated DEGs in *Mrtfb* cKO were associated with cytoskeletal protein binding, as well as a few of ancestor terms (*Figure 8J*).

Taken together, our hair cell transcriptome sequencing analysis revealed that *Srf* plays a vital role in activating genes involved in cell growth, defense response, and regulation of the actin cytoskeleton,

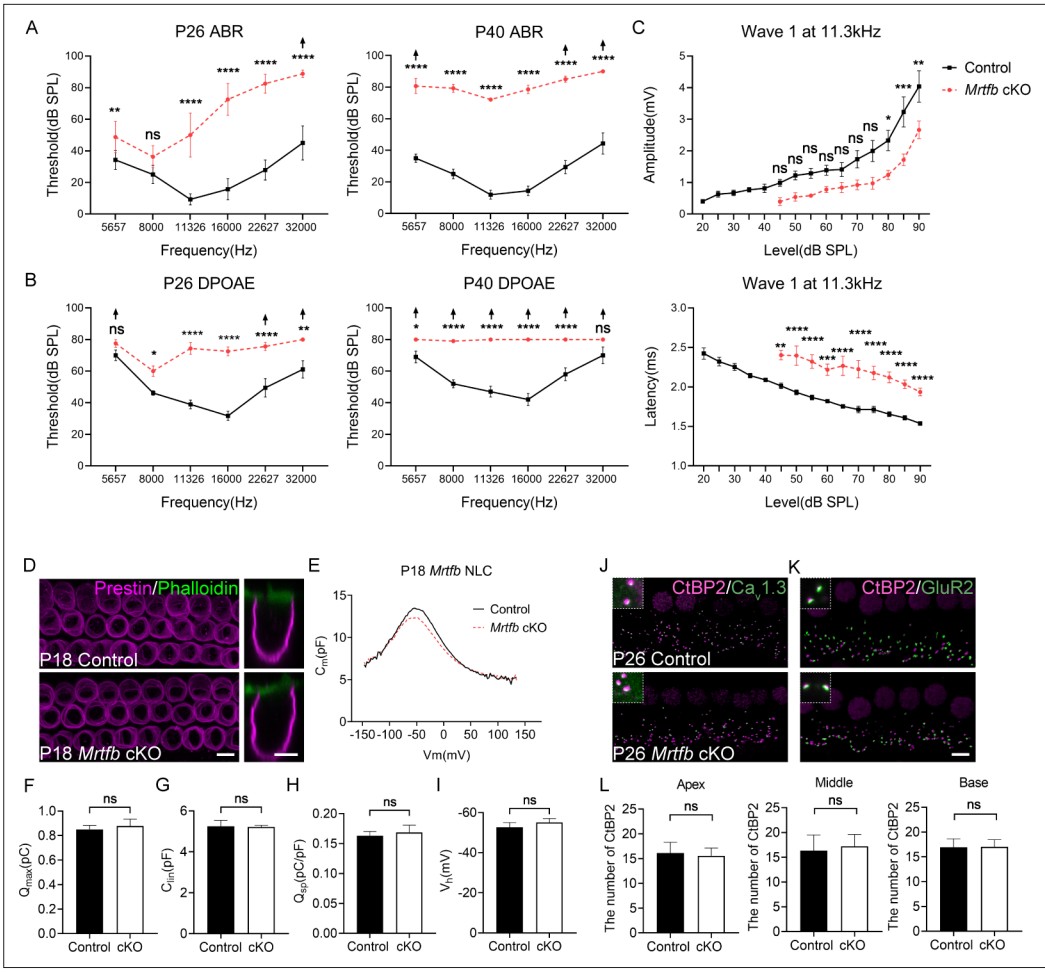

**Figure 7.** Early-onset and progressive hearing loss in *Mrtfb* cKO mice. (**A**) Auditory brainstem response (ABR) analysis demonstrates *Mrtfb* cKO mice exhibited early-onset and progressive hearing loss. (**B**) Distortion product otoacoustic emission (DPOAE) thresholds in *Mrtfb* cKO mice were significantly elevated at P26 and at P42 compared to control. (**C**) ABR wave1 amplitude (upper panel) and latency (lower panel) at 11.3 kHz in control and *Mrtfb* cKO mice at P26. Analyzed numbers (animals): P26, control (7), *Mrtfb* cKO (8). P42, control (8), *Mrtfb* cKO (7). Arrows indicate that at fixed frequency, no response was identified at the maximum output. Error bars indicate standard error of the mean (SEM), p values were derived from two-way analysis of variance (ANOVA) followed by Bonferroni post-test. (**D**) Prestin immunostaining in outer hair cells (OHCs) at P18, side views of OHCs are on the right. (**E**) Nonlinear capacitance (NLC) recordings of OHCs in control and *Mrtfb* cKO mice at P18. (**F–I**) Quantitative analysis of $Q_{max}$, $C_{lin}$, $Q_{sp}$, and $V_h$ showed no significant difference between control and *Mrtfb* cKO OHCs. Analyzed numbers (cells, animals): control (5, 4), *Mrtfb* cKO (5, 4). (**J, K**) CtBP2-Ca$_v$1.3 and CtBP2-GluR2 co-immunostaining in inner hair cells (IHCs). Higher magnification images are in the upper dotted box. (**L**) Comparison of IHC ribbon synapse numbers between control and *Mrtfb* cKO mice at different cochlear positions at P26. Analyzed numbers (cells, animals): apical turns, control (28, 5), *Mrtfb* cKO (28, 5). Middle turns, control (20, 6), *Mrtfb* cKO (26, 6). Basal turns, control (34, 6), *Mrtfb* cKO (32, 6). Scale bars in all panels represent 5 μm. Error bars indicate standard deviation (SD), p values were derived from two-tailed unpaired Student's *t*-test. ****p-value <0.0001, ***p-value <0.001, **p-value <0.01, and *p-value <0.05.

The online version of this article includes the following figure supplement(s) for figure 7:

**Figure supplement 1.** Hair cell numbers were not affected in *Mrtfb* cKO mice.

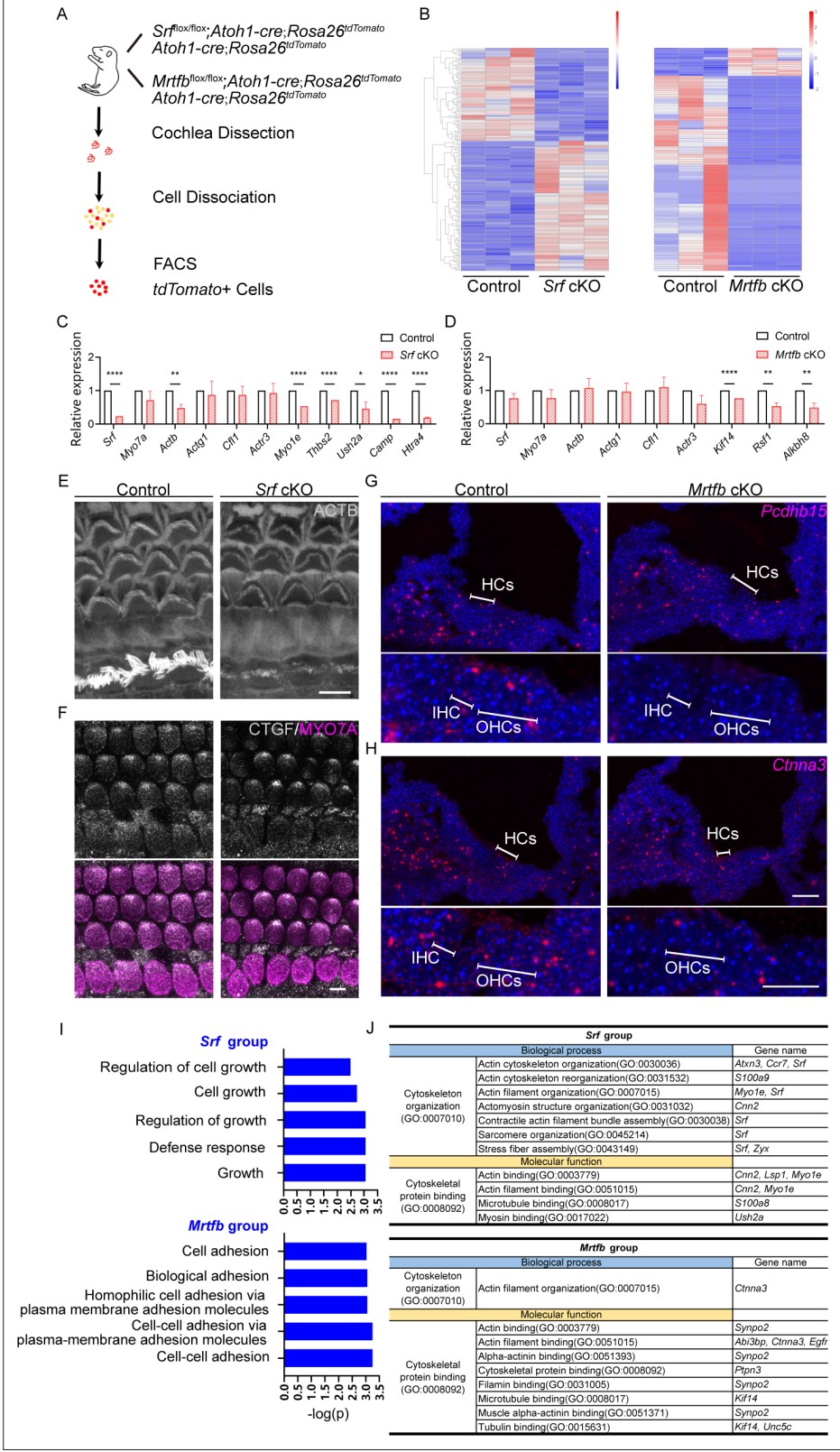

**Figure 8.** *Srf* cKO and *Mrtfb* cKO hair cell transcriptome analysis. (**A**) Diagram of fluorescence-activated cell sorting (FACS) purification strategy to isolate hair cells. The genotypes of mice used for cell sorting are on the right of the mouse cartoon. (**B**) The heatmaps of all the differentially expressed genes (DEGs) in *Srf* cKO and *Mrtfb* cKO ($p_{adj} < 0.05$ and $|log_2(fold change)| > 1$). Color-coded key bar indicates normalized expression values,

*Figure 8 continued on next page*

*Figure 8 continued*

relatively high expression levels are indicated by red colors, whereas blue colors represent lower expression levels. (**C, D**) RT-qPCR validation of the expression levels of genes identified in control, *Srf* cKO and *Mrtfb* cKO RNA-Seq, normalized to GAPDH. Error bars indicate standard error of the mean (SEM), p values were derived from one-way analysis of variance (ANOVA) followed by Bonferroni post-test, ****p value<0.0001, **p value<0.01, and *p value<0.05. (**E, F**) Reduced ACTB and CTGF immunostaining in *Srf* cKO HCs. Scale bars, 5 µm. (**G, H**) RNAscope in situ hybridizations analysis of *Pcdhb15* and *Ctnna3* expression in P2 frozen sections of cochleae from control and *Mrtfb* mutants. Nuclei were labeled with 4'-6-diamidino-2-phenylindole (DAPI). Scale bar, 20 µm. (**I**) Gene Ontology (GO) enrichment analysis (Biological Process) of down-regulated DEGs in *Srf* cKO and *Mrtfb* cKO. (**J**) Summary of functional clustering of down-regulated DEGs by *Srf* or *Mrtfb* deficiency, according to gene's GO annotations in Gene Ontology Resource.

The online version of this article includes the following figure supplement(s) for figure 8:

**Figure supplement 1.** Validation of RNA-sequencing results.

while *Mrtfb* functions independently of *Srf* and primarily directs gene expression in cell adhesions and regulation of actin cytoskeleton. These findings provide new insights into the mechanisms underlying hair cell development and maintenance.

## AAV-mediated rescue of *Cnn2* expression in vivo partially restores actin cytoskeleton dysfunction in *Srf* mutants

CNN2, an isoform of calponin proteins, which belong to a highly conserved family of actin-binding proteins, has been found in various cell types, such as smooth muscles, fibroblasts, endothelial cells, blood cells, and prostate cancer cells. In non-muscle cells, *Cnn2* plays a role in actin-based cellular activities, like cell migration and cell adhesion. In prostate cancer cells, *Cnn2* has been reported as an androgen-responsive *Srf* target gene that affects cell migration (*Hossain et al., 2005*; *Liu and Jin, 2016*; *Takahashi et al., 1988*; *Takahashi et al., 1986*). A motif matching SRF-binding CArG boxes has been identified 133 bp upstream of the transcriptional start site of *Cnn2*, and the binding of SRF to this site has been further confirmed by ChIP experiments (*Verone et al., 2013*). However, the functional significance of *Cnn2* in actin-based hair cells remains unknown.

In our transcriptomic analysis of *Srf* cKO hair cells, we observed a significant reduction in *Cnn2* expression (*Figure 8—figure supplement 1A*). RNAscope analysis revealed that *Cnn2* expression was highly enriched in the spiral ganglion neuron, stria vascularis, and spiral ligament, whereas low-level expression was also detected in the organ of Corti (*Figure 9—figure supplement 1A*).Immunocytochemistry showed that CNN2 was localized to the tips of row 1 stereocilia in IHCs and OHCs (*Figure 9A* and *Figure 9—figure supplement 1B, D*), and additional immunoreactivity was observed in the cuticular plate (*Figure 9B* and *Figure 9—figure supplement 1D*). The antibody used here recognizes an epitope located within the C-terminal variable region, which constitutes the main differences among the CNN isoforms. In contrast to control, the relative row 1 tip signal intensity decreased significantly for CNN2 in P6 *Srf* cKO IHC stereocilia (*Figure 9C*). A slightly reduced signal was also observed in the cuticular plate of *Srf* cKO IHCs and OHCs (*Figure 9B* and *Figure 9—figure supplement 1D*). These results suggest a key role of *Cnn2* downstream of *Srf* in the proper development of stereocilia and cuticular plate.

We then investigated whether the delivery of *Cnn2* using Anc80L65 in *Srf* cKO mice could rescue abnormal hair bundle and the cuticular plate structure. The Anc80L65 vector carrying the coding sequence of *Cnn2* followed by GFP reporter was delivered directly into the scala media of the basal turn of the cochlea in P0–P1 mice (*Figure 9D*). The transduction efficiency of Anc80L65 was visualized by GFP fluorescence. As illustrated in *Figure 9—figure supplement 1C*, almost all IHCs were GFP positive, while the majority of OHCs were transduced efficiently. Immunostaining using antibody against CNN2 showed that *Cnn2* delivered by Anc80L65 was highly expressed and specifically localized to the tips of row 1 stereocilia and the cuticular plate. Of note,transduced hair cell also showed row 2 stereocilia tip localization of CNN2 (*Figure 9E, F* and *Figure 9—figure supplement 1E*).

The relative F-actin intensity change of the cuticular plate was quantified and was found to be significantly increased in both IHCs and OHCs from *Srf* cKO mice injected with Anc80L65-Cnn2 compared to those injected with Anc80L65-GFP (*Figure 9G, H*). We also observed partial rescue of the length and width of row 1 stereocilia in IHCs (*Figure 9I, J*) and mild recovery of OHC stereocilia dimensions

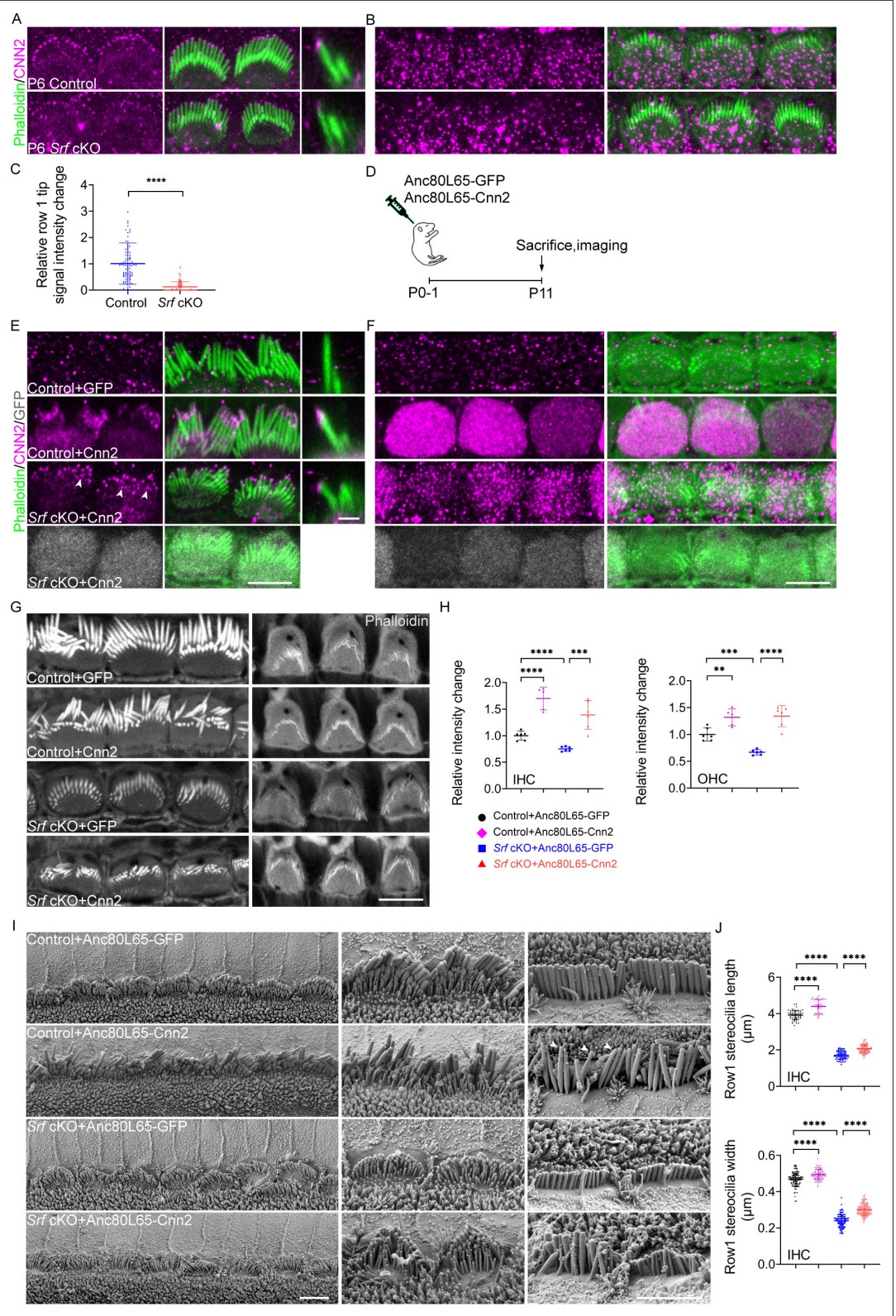

**Figure 9.** The injection of Anc80L65-Cnn2 partially restores the cuticular plate integrity and stereocilia morphology in *Srf* cKO mice. (**A, B**) CNN2 immunostaining in the stereocilia and cuticular plates of P6 inner hair cells (IHCs). (**C**) Quantitation of CNN2 reactivity in row 1 IHC stereocilia. Analyzed numbers (stereocilia, cells, animals): control (100, 10, 4), *Srf* cKO (110, 11, 4). (**D**) Depiction of the experimental paradigm for in vivo virus injection. (**E, F**) CNN2

*Figure 9 continued on next page*

*Figure 9 continued*

immunostainings in the stereocilia and cuticular plates of P11 control mice injected with Anc80L65-GFP or Anc80L65-Cnn2, and *Srf* cKO injected with Anc80L65-Cnn2.The bottom panels in E and all panels in F were taken at the level of the cuticular plate. The GFP expression in IHCs of *Srf* cKO injected with Anc80L65-Cnn2 indicate the efficiency of *Cnn2* delivery. Arrowheads indicate CNN2 at row 2 stereocilia tips. (**G**) Phalloidin staining in the cuticular plates of Anc80L65-GFP-injected control, Anc80L65-Cnn2-injected control, Anc80L65-GFP-injected *Srf* cKO, and Anc80L65-Cnn2-injected *Srf* cKO hair cells at P11. (**H**) Quantification of phalloidin reactivity in the cuticular plates of P12 IHCs and outer hair cells (OHCs) from Anc80L65-GFP-injected control, Anc80L65-Cnn2-injected control, Anc80L65-GFP-injected *Srf* cKO, and Anc80L65-Cnn2-injected *Srf* cKO mice. Analyzed numbers (cells, animals): IHCs, Anc80L65-GFP-injected control (52, 5), Anc80L65-Cnn2-injected control (47, 5), Anc80L65-GFP-injected *Srf* cKO (49, 5), Anc80L65-Cnn2-injected *Srf* cKO (61, 6). OHCs, Anc80L65-GFP-injected control (194, 5), Anc80L65-Cnn2-injected control (147, 5), Anc80L65-GFP-injected *Srf* cKO (150, 5), Anc80L65-Cnn2-injected *Srf* cKO (204, 6). (**I**) Representative scanning electron microscopy (SEM) images of P11 apical IHCs of Anc80L65-GFP-injected control, Anc80L65-Cnn2-injected control, Anc80L65-GFP-injected *Srf* cKO, and Anc80L65-Cnn2-injected *Srf* cKO. Arrowheads indicate abnormal thin stereocilia tips. (**J**) SEM-based quantification of row 1 stereocilia length and width of P11 apical IHCs. Analyzed numbers (stereocilia, cells, animals): lengths, Anc80L65-GFP-injected control (52, 16, 4), Anc80L65-Cnn2-injected control (44, 13, 5), Anc80L65-GFP-injected *Srf* cKO (70, 20, 5), Anc80L65-Cnn2-injected *Srf* cKO (99, 30, 6). Widths, Anc80L65-GFP-injected control (107, 20, 4), Anc80L65-Cnn2-injected control (75, 20, 5), Anc80L65-GFP-injected *Srf* cKO (121, 25, 5), Anc80L65-Cnn2-injected *Srf* cKO (130, 30, 6). In E, the scale bar for side-view represents 1 μm and scale bars in other panels represent 5 μm. Error bars indicate standard deviation (SD), p values were derived from two-tailed unpaired Student's *t*-test, ****p value<0.0001, ***p value<0.001, and **p value<0.01.

The online version of this article includes the following figure supplement(s) for figure 9:

**Figure supplement 1.** The injection of AAV-Anc80L65-Cnn2 partially restores outer hair cell (OHC) stereocilia morphology in *Srf* cKO mice.

---

(*Figure 9—figure supplement 1F, G*). Notably, the control mice injected with Anc80L65-Cnn2 showed more F-actin intensity of the cuticular plate compared to the hair cells with Anc80L65-GFP (*Figure 9G, H*). Meanwhile, we observed slightly elongated and widened row 1 stereocilia in both control IHCs and OHCs with Anc80L65-Cnn2 (*Figure 9I, J* and *Figure 9—figure supplement 1F, G*), suggesting the role of *Cnn2* in controlling stereocilia actin polymerization and the F-actin organization of the cuticular plate. As indicated by arrowheads in *Figure 9I*, the overexpression of CNN2 in control hair cells led to elongated but tapering stereocilia top, which may be caused by excessive expression of CNN2 along the upper part of the stereocilia (*Figure 9E*). Similarly, the row 3 and 4 IHC stereocilia seemed elongated in *Srf* cKO with Anc80L65-Cnn2 transduction, possibly due to ectopic overexpression of CNN2 in the cuticular plate and lower rows of hair bundle (*Figure 9E, F*). In conclusion, *Cnn2*, as a downstream target and an effector of *Srf*, regulates the development of stereocilia dimensions and F-actin network integrity of thecuticular plate in hair cells.

## Discussion

The actin cytoskeleton has been recognized as a fundamental structure that plays a crucial role in normal hearing, and abnormal functioning of actin cytoskeleton-related proteins can cause hearing loss or deafness. However, the mechanisms of transcriptional controlling of actin cytoskeleton genes in hair cells are not yet well understood. The MRTF–SRF axis is a key signaling pathway that regulates the expression of genes participating in actin cytoskeleton activities, and has been studied extensively in a broad range of tissues and cell types. In this study, as illustrated in *Figure 10*, we investigated the importance of *Srf* and *Mrtf*s in hair cell actin-based structures by targeted inactivation of *Srf*, *Mrtfa* and *Mrtfb*, respectively. Our results showed that both *Srf* and *Mrtfb* are indispensable for the proper development of stereocilia dimensions, represented by the altered expression patterns of key stereocilia tip proteins in mutants, and maintaining the integrity of the cuticular plates. As expected, *Srf* and *Mrtfb* cKO hair cells exhibited a phenotypic similarity in reduced length and width of the stereocilia, and less dense F-actin network of the cuticular plate, suggesting a potential contribution of the MRTFBSRF axis to overlapping phenotypes. However, RNA-sequencing of hair cells and functional clustering analysis of down-regulated DEGs revealed unanticipatedly distinct profiles of genes regulated by *Srf* and *Mrtfb*, highly suggesting different transcriptional regulation mechanisms of actin

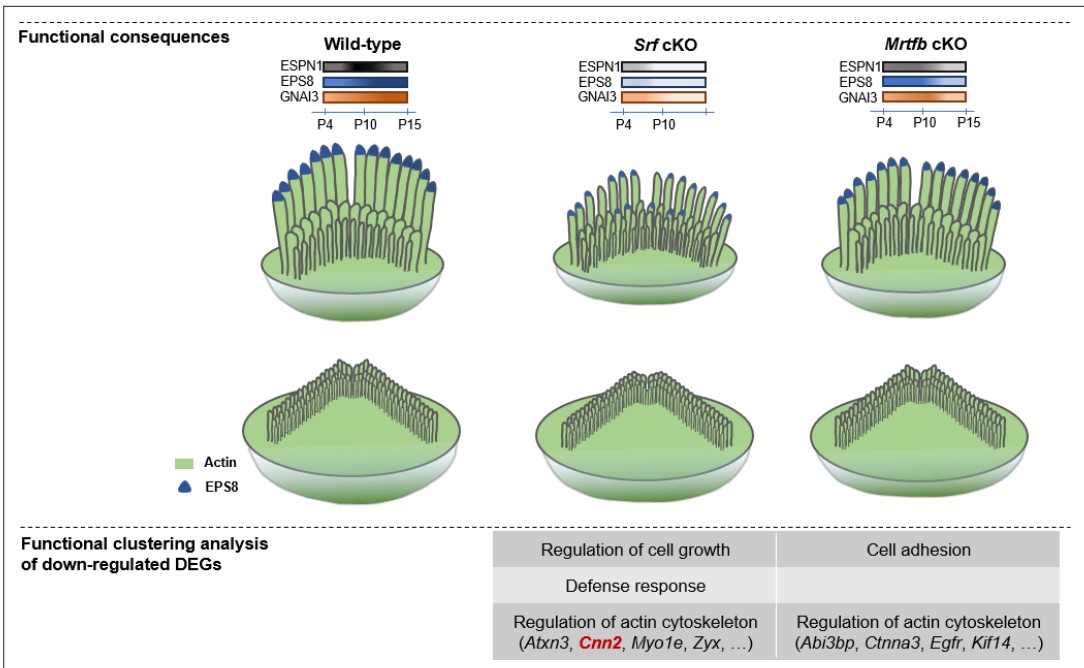

**Figure 10.** An illustration model of differential regulation of actin cytoskeleton mediated by SRF and MRTFB in hair cells.

cytoskeleton mediated by *Srf* and *Mrtfb*. Based on DEGs analysis, we further provided evidence suggesting the involvement of *Cnn2*, one of the *Srf* downstream targets, in regulating the hair bundle morphology and cuticular plate F-actin network organization. Notably, FACS-based RNA-sequencing is a powerful approach to resolve molecular differences in specific cell types between control and mutants, enabling the identification of functionally important genes for hair cells that have not been identified by genetics, such as actin cytoskeleton genes that express ubiquitously, but act essential roles in actin-based cytoskeleton structures of hair cells.

Recently, emerging evidence suggests that MRTFs can also direct transcription of genes involved in multiple aspects of cell migration by cooperating with other transcription factors besides *Srf*. For example, the interaction between MRTFA and STAT3 promotes the migration of breast cancer cells (*Liao et al., 2014*; *Xing et al., 2015*), and the crosstalk of MRTF-YAP is required for cancer cell migration and metastasis (*Kim et al., 2017*). Additionally, *Mrtf*s can promote transcription of genes via its own SAP domain, as shown in Irem Gurbuz's study that *Mrtfa* stimulates a breast cancer-specific set of genes involved in cell proliferation and cell motility (*Gurbuz et al., 2014*). Actin-binding proteins profilin isoforms *Pfn1* and *Pfn2* are also regulated by *Mrtf*s utilizing SAP domain-mediated transcriptional activity (*Joy et al., 2017*). In this study, hair cell-specific deletion of *Mrtfb* yielded a *Srf*-like phenotype outwardly, such as defects in the stereocilia dimensions and the cuticular plate integrity, but the lack of *Srf* or lack of *Mrtfb* had differential effects on hair cell transcriptome. These results highlight the complexity and context- specific regulation of actin cytoskeleton activities mediated by *Srf* and *Mrtf*s, and shed light on the potential of *Mrtfb* to regulate actin cytoskeleton in an *Srf*-independent manner. We summarized recent studies and screened the reported genes potentially regulated by *Mrtf*s in an *Srf*-independent mode (SAP, STAT3, or YAP dependent) in our hair cell RNA-sequencing dataset. Only *Nox4* and *Kif14* showed significantly differential expression in *Mrtfb* mutants (*Figure 8— figure supplement 1B*), and the reduction of *Kif14* was also confirmed by RT-qPCR (*Figure 8D*). It is appealing to further investigate the mechanisms underlying *Mrtfb*-mediated regulation, as well as the interaction between *Mrtfb* and its transcription factors in the context of hair cell.

As previously noted, hair cells lacking *Srf* or *Mrtfb* displayed some common features such as decreased stereocilia length and width, and a less dense meshwork of F-actin forming the cuticular plate.However, the impairments were more severe in *Srf*-deficient hair cells compared to *Mrtfb* mutants, which showed significantly delayed microvillar regression and kinocilium degeneration.

Analysis of DEGs revealed a reduction in *Egr1*, an immediate early gene and well-known transcriptional target of *Srf*, in *Srf* cKO hair cells. Various studies have established *Egr1* as a regulator of hematopoietic cell differentiation and maturation (*Kulkarni, 2022*). *Ctgf* which was also significantly reduced in *Srf* cKO hair cells, was found to be enriched in cell growth-associated GO terms. Previous study showed that *Mrtfa* regulated microvessel growth via *Ccn1* and maturation via *Ctgf* to mediate functional improvement of ischemic muscle tissue (*Hinkel et al., 2014*). These findings suggest that hair cell maturation may be affected by *Srf* deficiency, and delayed maturation may contribute to more severe defects in the actin cytoskeleton organization of stereocilia and cuticular plate, as well as microvillar regression and kinocilium degeneration.

In this study, an AAV vector containing wild-type *Cnn2* cDNA was generated and injected into the cochleae of early postnatal *Srf* cKO mice. Immunofluorescence staining revealed a much stronger CNN2 signal, primarily restricted to the cuticular plate and the tips of stereocilia, unlike endogenous CNN2 expression in age-matched control mice injected with Anc80L65-GFP (*Figure 9E, F*). Upon Anc80L65-Cnn2 injection, the raised apical plasma membrane of cuticular plate was barely seen in *Srf* cKO mice (*Figure 9—figure supplement 1F*). In terms of F-actin intensity of the cuticular plate, *Srf* cKO hair cells appeared to be fully restored by *Cnn2* delivery, with even stronger F-actin intensity compared to control. However, hair cell stereocilia dimensions defects were only partially restored.

The partial rescue of stereocilia dimensions observed upon AAV-mediated delivery of *Cnn2* in *Srf* cKO mice could be attributed to several reasons. Firstly, immunofluorescence staining results indicated a transient expression pattern of CNN2 during stereocilia development. The CNN2 staining can be detected as early as P0, and it began to accumulate at row 1 tip, reaching a peak at P6, then decreased at P10. (*Figure 9—figure supplement 1B*). Alterations in CNN2 expression levels were also observed in the cuticular plate of P6 and P11 hair cells (*Figure 9B, F*), suggesting a critical window of gene delivery for successful recovery of stereocilia dimensions. Recent advances in utero gene transfer to the inner ears have been reported (*Gubbels et al., 2008*; *Kim et al., 2016*; *Miwa et al., 2013*), especially AAV-mediated prenatal gene transduction (*Bedrosian et al., 2006*; *Hu et al., 2020*), indicating the potential use of this approach to initiate the *Cnn2* overexpression earlier during stereocilia development. Secondly, *Cnn2* has been reported to regulate cell motility differently in various cell types and biological processes. For example, the macrophages of *Cnn2* knockout mice showed faster migration than that of control mice (*Huang et al., 2008*). In Tang's in vitro study, human umbilical vein endothelial cell migration was enhanced by *Cnn2* overexpression and inhibited by *Cnn2* knockdown (*Tang et al., 2006*). These discrepancies suggested that the precise level of *Cnn2* expression required for maintaining actin cytoskeleton activities, such as cell migration, might also apply in hair bundle development and warrants further investigation. Thirdly, our differential gene expression analysis identified 79 down-regulated genes in *Srf* cKO hair cells, including *Cnn2*. Using the Gene Transcription Regulation Database (GTRD, v20.06), we extracted publicly available mouse ChIP-seq experiments information on *Srf* (Q9JM73), and screened the regions [−1000,+100] bp around the transcriptional start site of each gene on the list of 79 down-regulated *Srf* targets for *Srf*-binding sites (data not shown). Except for *Cnn2*, which had the most *Srf*-binding sites, several candidate target genes implicated in the regulation of actin cytoskeleton, such as *Lsp1* and *Zyxin*, were also identified. The leukocyte-specific protein 1, *Lsp1* is found in several cells types, such as neutrophils and endothelial (*Li et al., 1995*; *Liu et al., 2005*), and interacts with F-actin through villin headpiece-like sequence (*Klein et al., 1990*; *Zhang et al., 2001*). Further evidence suggests its function in regulating actin cytoskeleton remodeling (*Maxeiner et al., 2015*). The LIM-domain protein ZYXIN has been reported to regulate actin polymerization and play an important role in the organization of actin cytoskeleton. These findings suggest that the precise control of stereocilia dimensions and F-actin network integrity of cuticular plate might be governed by multiple candidate genes cooperatively (*Fradelizi et al., 2001*; *Li et al., 2004*).

It was surprising that late embryonic deletion of SRF, which is considered a master regulator of the actin cytoskeletal network (*Miano et al., 2007*), only mildly affected hair cell development. Instead, based on the observed rapid postnatal degeneration of the stereocilia and cuticular plate, it appears that SRF is equally or more important for the maintenance of the hair cell's F-actin structures. We believe that this finding is important, as it will provide the foundation for future research on the role of SRF and its downstream components in the long-term maintenance of the hair bundle, with potential implications for progressive types of hearing loss.

# Materials and methods

## Animals

Experimental mice were housed with a 12:12hr light:dark cycle in standard laboratory cage in a temperature and humidity-controlled vivarium. Both male and female mice were used in all experiments. *Srf*<sup>flox/+</sup> mouse was a gift from Dr. Eric Olson (University of Texas Southwestern Medical Center, USA), and *Atoh1-Cre* mouse line was kindly provided by Dr. Lin Gan (University of Rochester, USA). B6.*Cg-Gt(ROSA)26Sor*<sup>tm9(CAG-tdTomato)Hze</sup>/J (tdTomato) reporter mice (Stock No. 007909, C57BL/6J) were ordered from Jackson Laboratories. For *Mrtfa* and *Mrtfb* floxed lines (C57BL/6J), the target sequence was chosen using the CRISPR Design bioinformatics tool developed by Feng Zhang's lab at the Massachusetts Institute of Technology (https://zlab.squarespace.com/). The conditional *Mrtfa* allele contains two loxP sites flanking exon 3, and the target sequences used were: TCTAACCAAGCCGTTAGTCC, GCAGTCCAGTGGGGGCACTA. The exon 3 to exon 17 of *Mrtfb* was targeted and flanked by two loxP sites, and the target sequences used were: CCATGGGCTAAATTACACCG, CAGGTAAAGTATCATCCTAC. The floxing *Mrtfa* and *Mrtfb* lines were generated and obtained from GemPharmatech Co., Ltd, Nanjing, China. *Srf*<sup>fl/fl</sup>; *Atoh1-Cre* crosses were maintained in the C57BL/6J and 129S6 mixed background. *Mrtfa*<sup>fl/fl</sup>; *Atoh1-Cre* and *Mrtfb*<sup>fl/fl</sup>; *Atoh1-Cre* were on a mixed background dominated by C57BL/6J (bred five generations to the C57BL/6J background). The genotypes used in this study are as follows: *Srf*<sup>fl/fl</sup> (control), *Srf*<sup>fl/fl</sup>; *Atoh1-Cre* (*Srf* cKO), *Mrtfa*<sup>fl/fl</sup> (control), *Mrtfa*<sup>fl/fl</sup>; *Atoh1-Cre* (*Mrtfa* cKO), *Mrtfb*<sup>fl/fl</sup> (control), *Mrtfb*<sup>fl/fl</sup>; *Atoh1-Cre* (*Mrtfb* cKO). Genotyping primers for all mutants are listed in ***Supplementary file 1***.

## Immunocytochemistry

Whole inner ears were fixed for 30 min in 4% paraformaldehyde at room temperature. Adult cochleae were decalcified with 4.13% Ethylenediaminetetraacetic acid (EDTA) before dissection. After dissection, tissues were blocked for 1 hr with 1% bovine serum albumin, 3% normal donkey serum, and 0.2% saponin in phosphate-buffered saline (PBS; blocking buffer), and incubated overnight at 4°C with primary antibodies in blocking buffer. Tissues were then washed with PBS and incubated with secondary antibodies (Alexa 488, Alexa 555, Alexa 647-conjugated donkey anti-rabbit IgG, donkey anti-mouse IgG, donkey anti-goat IgG, Invitrogen) and phalloidin-Alexa 488 (Invitrogen) in blocking buffer for 1–2 hr. Finally, organs were washed with PBS and mounted in Vectashield (Vector Laboratories). For immunostaining of ESPN1, EPS8, GNAI3, and FSCN2 proteins, tissues were fixed in 4% paraformaldehyde for 15–20 min at room temperature. For measurements of stereocilia length visualized by phalloidin staining, tissues were fixed in 4% paraformaldehyde for 12–24 hr at 4°C. After dissection, tissues were permeabilized in blocking buffer for 1 hr at room temperature, and then incubated with phalloidin-Alexa 488 (Invitrogen) in blocking buffer for 3–4 hr at room temperature. Samples were imaged using Zeiss LSM880 and Leica SP8 confocal microscopes.

For the analysis of the phalloidin reactivity in the cuticular plates, hair cells containing upright hair bundles were used for statistics. summed Z-projections were made that included the whole cuticular plate. Regions of interest (ROIs) were selected using circles that encompassed the center area of the cuticular plate. ROIs were measured using ImageJ Fiji's Measure function and measurements were also made outside the hair cells and sensory epithelium to determine the background. The phalloidin reactivity in one cuticular plate was (area) * (mean gray value), minus background signal. The phalloidin reactivity measurement method was described previously (***Du et al., 2019***). The measurement of cuticular plate volume was performed using Imaris version 9.9.0 (Oxford Instruments, Abingdon, UK) by manually selecting the area at the top of the cuticular plate where the phalloidin stain begins to appear until disappears at the bottom for 3D reconstruction.

For the analysis of the length of stereocilia, images containing upright hair bundles were selected for generating x–z reslices of stereocilia using ImageJ Fiji's Reslice function. Reslices with views of central row 1 stereocilia in bundles were chosen. A vertical line from the top of a stereocilium to the point of insertion in the cuticular plate was drawn to measure a single stereocilium length. Stereocilia of HCs in the same region of the middle turn (20–24 kHz) and the basal turn (32–36 kHz) were measured.

For the analysis of the row 1 tip signal intensity of tip proteins, isolated Z-projections were made for row 1 tips and ROIs were selected at the top 1–2 μm of each tip using circles. ROIs covered most

of each tip and 10 or more row 1 tips per hair bundle were measured using ImageJ Fiji's Measure function. Measurements were also made outside the hair cells and sensory epithelium to determine the background. The tip signal intensity in a tip was (area) * (mean gray value) * (number of stacks used for the average projection), minus the total background signal. The tip signal intensity below the background was assigned a value of 0 for data calculation. For each protein staining, two to three images were acquired from 1 to 2 mice for each genotype at different ages per experiment, and experiments were repeated at least twice. Stereocilia length and tip signal intensity measurement methods have been reported previously (*Krey et al., 2020*).

The following antibodies were used in this study: rabbit anti-Myo7a (111501, Proteus BioSciences), rabbit anti-Srf (16821-1-AP, Proteintech), mouse anti-Lmo7 (B-7, sc-376807, Santa Cruz Biotechnology), rabbit anti-Triobp (16124-1-AP, Proteintech), mouse anti-Mrtfa (G-7, sc-398675, Santa Cruz Biotechnology), mouse anti-MKL2/Mrtfb (CL1546, MA5-24628, Thermo Fisher Scientific), mouse anti-Espn (611656, BD Biosciences), mouse anti-Eps8 (610143, BD Biosciences), rabbit anti-Gnai3 (G4040-1VL, Sigma-Aldrich), goat anti-Fscn2 (EB08002, Everest Biotech), rabbit anti-Prestin (EPR22715-53,ab242128, abcam), mouse anti-Ctbp2 (612044, BD Biosciences), mouse anti-glutamate receptor (556341, BD Biosciences), rabbit anti-Ca$_v$1.3 (ACC-005, Alomone Labs), mouse anti- Acetylated tubulin (T6793-100UL, Sigma-Aldrich), mouse anti-Ctgf (E-5, sc-365970, Santa Cruz Biotechnology), mouse anti-CD45 (AF114, R&D Systems), mouse anti-beta Actin (AC-15, ab6277, abcam), and rabbit anti-Cnn2 (21073-1-AP, Proteintech).

## Scanning electron microscopy

Adult mice were anesthetized and perfused intracardially with 2.5% glutaraldehyde + 2% paraformaldehyde. The cochleae were dissected and perfused through the round window with the fixative (2.5% glutaraldehyde, in 0.1 M cacodylate buffer, with 3 mM CaCl$_2$). Cochleae were then incubated in fixative overnight at 4°C. For neonatal mice, the samples were dissected and treated with fixative immediately. The cochleae were decalcified with 4.13% EDTA for several days at room temperature, and then further dissected to expose the sensory epithelium. Samples were post-fixed according to the thiocarbohydrazide-osmium protocol (*Davies and Forge, 1987*). Briefly, samples were incubated in 1% OsO$_4$ for 1 hr, washed in distilled water, and then incubated in 1% thiocarbohydrazide for 20 min. This incubation process was repeated twice followed by incubation again in 1% OsO$_4$ for half an hour. Samples were then processed to dehydration with graded ethanol, critical-point dry and mounting on stubs. After sputter coating with platinum using Quorum Q150T ES, they were imaged on a Zeiss GeminiSEM 300 using the secondary electron detector. To determine the length of individual stereocilia in the highest row of a hair bundle, we obtained stereopairs scanning electron micrographs of the same hair bundle. The stereopairs of images were captured with 10° of tilt. The stereocilia length was quantified using previously described methods (*Thiede et al., 2014*; *Vélez-Ortega et al., 2017*). The widest part of a single stereocilium was measured as stereocilium width. Hair bundles of HCs in the basal turn (30–32 kHz) were selected for SEM imaging and data analysis.

## Transmission electron microscopy

Adult mice were anesthetized and perfused intracardially with 2.5% glutaraldehyde + 2% paraformaldehyde. The cochleae were dissected and perfused through the round window with the fixative. For neonatal mice, the samples were dissected and treated with fixative immediately. Cochleae were incubated in fixative overnight at 4°C and then decalcified in 4.13% EDTA pH 7.3 in 2% paraformaldehyde, for 1 week at room temperature. The cochleae were post-fixed in 1% OsO$_4$–1.5% ferricyanide in 0.1 M PB, then stained with 4% uranyl acetate. The samples were dehydrated in graded series of ethanol, acetone oxime, and 100% Epoxy Embedding Medium (EPON). Samples were sectioned at 100 nm thickness onto 200 mesh copper grids and contrast stained with lead citrate and 4% uranyl acetate. Imaging was conducted using a Hitachi HT7800 transmission electron microscope.

## Auditory brainstem response

ABRs were recorded at P26 and P42. The measurements were performed in a sound-attenuating chamber (Shanghai Shino Acoustic Equipment Co., Ltd). Animals were anesthetized with intraperitoneal injection of zoletil (50 mg/kg) and xylazine (20 mg/kg), and then kept on a heating pad (Homeothermic Monitoring System, Harvard Apparatus). ABRs were recorded by three subdermal needle

electrodes. The active electrode was placed at the vertex of the skull, the reference electrode over the mastoid area of the right ear and the ground electrode on the shoulder of the left side. The free-field sound stimuli were delivered using a speaker (MF1, Tucker-Davis Technologies, Inc, Alachua, FL, USA) placed in front of the vertex of animal about 10 cm. Sound signals were generated by an acoustic stimulation system (Tucker-Davis Technologies), and data were acquired with the software BioSigRZ (Tucker-Davis Technologies). Responses were amplified ×5000 and filtered at 0.03–5 kHz, and thresholds were measured at 5.6, 8, 11.3, 16, 22.6, and 32 kHz. Stimulus intensity was decreased in 5 dB steps until two response waveforms could no longer be identified. Hearing threshold was defined as the lowest sound pressure level that elicited an appropriate ABR response.

## Distortion product otoacoustic emissions

DPOAE was recorded at P26 and P42. Mice were anesthetized and kept on a heating pad. Two EC1 electrostatic speakers driven by Tucker-Davis Technologies (TDT) RZ6 system through separate channels generated two stimulus frequencies ($f1$ and $f2$). The speakers were connected to the right ear of animals using a close-field component. DPOAE recordings were collected with constant $f2$ frequencies from 5 to 32 kHz, with $f1$ swept with the frequency ratio of $f2/f1$ constant at 1.2. The level of L1 was equal to L2. L2 was swept in 5 dB increments from 20 to 80 dB SPL for each frequency. The acoustic waveform was collected using BioSigRZ software. DPOAE threshold was defined as the L2 producing a DPOAE that exceeded the mean noise floor by at least 3 dB. For each frequency, the mean noise floor intensity was under 0 dB SPL.

## Hair cell counts

Mice were killed after ABR and DPOAE tests at P26 and P42. Cochleae were harvested and fixed in 4% paraformaldehyde for 24 hr and decalcified for 2 days in EDTA solution. Apical, middle, and basal turns of the cochlear sensory epithelium were dissected. The presence of hair cells was labeled with MYO7A immunoreactivity. Confocal microscopy images were analyzed using ImageJ. Hair cells were counted in at least 200 μm widths of the apical turn (8–11 kHz), middle turn (20–24 kHz), and basal turn (32–34 kHz) in each cochlear sensory epithelium.

## Whole-cell patch-clamp recording

All the patch-clamp experiments were performed at room temperature. The cochleae explants were dissected out from P18 mice. Individual first row OHCs were isolated from apical turns of the cochlea and bathed in extracellular solutions containing the following (in mM): 132 NaCl, 2 CaCl$_2$, 2 MgCl$_2$, and 10 4-(2-Hydroxyethyl)piperazine-1-ethanesulfonic acid (HEPES) (pH 7.2–7.3 with NaOH, osmolality 300 mOsm with D-glucose). The intracellular solution was the same as extracellular solution, except for the addition of 10 mM Ethylene-bis(oxyethylenenitrilo)tetraacetic acid (EGTA). The recording site corresponds to the 8–11 kHz region. For data collection, an axon 200B amplifier (Molecular Devices, Sunnyvale, CA) was used. Continuous high resolution (2.56 ms sampling) two-sine stimulus protocol (10 mV peak at both 390.6 and 781.2 Hz) superimposed onto a 300-ms voltage ramp (from+160 to −160 mV) was used for nonlinear capacitance measurement. Data were acquired and analyzed using jClamp (SciSoft Co., Ridgefield, CT; https://www.scisoftco.com/).

Capacitance data were fit to the first derivative of two-state Boltzmann function.

$$C_m = Q_{max} \frac{ze}{kT} \frac{b}{(1+b)^2} + C_{lin}$$

where

$$b = \exp(\frac{-ze(V_m - V_{pkCm})}{kT})$$

$Q_{max}$ is the maximum nonlinear charge, $C_{lin}$ is linear membrane capacitance, $V_{pkCm}$ ($V_h$) is voltage at peak capacitance, $V_m$ is membrane potential, $z$ is valence, $e$ is electron charge, $k$ is Boltzmann's constant, and $T$ is absolute temperature.

## Cochlear hair cell FACS

Cochlear sensory epithelia were dissected from *Srf*$^{fl/fl}$; *Atoh1-Cre; tdTomato*, *Mrtfb*$^{fl/fl}$; *Atoh1-Cre; tdTomato*, and *Atoh1-Cre; tdTomato* mice at P2. 8–20 epithelia were collected for each group. The collected cochlear sensory epithelia were incubated for 20 min at 37°C in an enzymatic solution containing papain (LK003150, Worthington Biochemical Corp) and dissociated by triturating with a pipette. The digestion was stopped by the addition of ice-cold ovomucoid in bovine serum albumin solution. The dissociated cells were spun at 300 × *g* for 5 min. After removing the supernatant, cells were resuspended with Dulbecco's phosphate-buffered saline (DPBS). The cells were filtered through a cell strainer with 35 µm mesh to eliminate cell clumps and debris. The cells were then sorted on BD Influx cell sorter with a 100-µm nozzle (BD Biosciences, San Jose, CA) using the channel for tdTomato to collect positive cells into DPBS. Hair cells, which comprised 1.2–1.8% of viable cells from each batch, were collected using tdTomato fluorescence signal. Approximately $6 \times 10^3$ to $1 \times 10^4$ tdTomato-positive cells were collected for each sample collection. After capturing single hair cells, cells were centrifuged for 5 min at 300 × *g* in 4°C. The supernatant was removed as much as possible before freezing the cells at −80°C.

## RNA-Seq

Total RNA was extracted from sorted hair cells using the RNAprep Pure Micro Kit (DP420, Tiangen Biotech) according to the manufacturer's instructions. RNA integrity was assessed using the RNA Nano 6000 Assay Kit of the Bioanalyzer 2100 system (Agilent Technologies, CA, USA). cDNA was synthesized using SMARTer Ultra Low RNA Kit for Illumina Sequencing (634936, Takara Bio). The DNA libraries were constructed using TruePrep DNA Library Prep Kit V2 for Illumina (TD503-02, Vazyme) following the manufacturer's instructions and quantified by Agilent Bioanalyzer 2100 system. RNA-Seq was performed on the Illumina NovaSeq 6000 system. After sequencing, clean reads were aligned to the mouse GRCm39 genome using Hisat2 (v2.0.5). Gene-level read counts were calculated using FeatureCounts v1.5.0-p3. The read counts of genes were normalized data of the Fragments Per Kilobase of transcript per million fragments mapped (FPKM). Differential expression analysis of genes was performed using the DESeq2 R package (1.20.0). The DEGs were determined by comparing *Srf* cKO to control, and *Mrtfb* cKO to control, respectively, with $p_{adj} < 0.05$ and $|\log_2(\text{fold change})| > 1$. GO enrichment analysis of DEGs was implemented by the clusterProfiler R package, in which gene length bias was corrected. The total RNA extraction, library construction, and sequencing were completed by Novogene (Novogene Bioinformatics Technology, Beijing, China).

## RT-qPCR

The cDNA of sorted tdTomato-positive hair cells was synthesized using PrimeScript RT reagent Kit with gDNA Eraser (Perfect Real Time, RR047A, Takara Bio). Synthesized cDNA was subsequently mixed with qPCR primers and TB Green Premix Ex Taq (Tli RNase H Plus, RR420A, Takara Bio). Real-time PCR quantification was performed using Cobas z 480 analyzer (Roche Diagnostics GmBH). The threshold cycle (Ct) number for each sample was determined in triplicate, and the experiments were repeated with three individual sets of samples. Relative gene expression levels were calculated using the ΔΔCt method, and *Gapdh* was used as reference gene. RT-qPCR was performed with paired primers listed in **Supplementary file 1**.

## RNAscope

Cochlear samples from neonatal mice were fixed in freshly prepared 4% paraformaldehyde for 24 hr at 4°C, washed in PBS, dehydrated using a sucrose gradient, and then embedded in O.C.T. Compound (4538, Sakura Finetek) for cryosectioning. Samples were sectioned at 12 µm thickness and mounted on SuperFrost Plus slides (Fisher). The slides were post-fixed with 4% paraformaldehyde at 4°C for 15 min, soaked in a graded ethanol series, and subjected to RNAscope staining using the RNAscope 2.5HD Red Assay (322350, ACD) and RNAscope Multiplex Fluorescent Reagent Kit v2 (323100, ACD) according to the manufacturer's standard protocol. The following RNAscope probes used in the experiment were designed by and ordered from ACD: mm-*Ctnna3* (516621), mm-*Pcdhb15* (544991), mm-*Mrtfa* (567821), mm-*Mrtfb* (404721), and mm-*Cnn2* (1231301-C1).

## AAV production

Anc80L65-CMV-GFP and Anc80L65-CMV-mouse Cnn2-P2A-GFP were purchased from Vigene Biosciences (Shandong, China). The titers of Anc80L65-CMV-GFP and Anc80L65-CMV-mouse Cnn2-P2A-GFP were $4.7 \times 10^{13}$ and $3.7 \times 10^{13}$ genome copies/ml, respectively.

## In vivo AAV gene delivery

$Srf^{fl/+}$, $Srf^{fl/fl}$, and $Srf^{fl/fl}$; $Atoh1$-$Cre$ pups were anesthetized by hypothermia on ice at P0–P1. A preauricular incision was made to expose the otic bulla. A total of 0.5 µl of each virus per cochlea were delivered into the scala media using a glass micropipette (WPI, Sarasota, FL) held by a Nanoliter 2000 micromanipulator (WPI). The injection site was the basal turn of the cochlea. The release speed was controlled by MICRO-2T microinjection controller (WPI) at 129 nl/min.

## Statistical analysis

Statistical analysis was performed using GraphPad Prism 8 (La Jolla, CA). One-way analysis of variance (ANOVA) followed by Bonferroni correction was used for RT-qPCR data, two-way ANOVA followed by Bonferroni post-test was used for ABR and DPOAE analyses, and two-tailed unpaired Student's $t$-test was used for other comparisons between two groups. p-values <0.05 were considered statistically significant. Asterisks in all figures indicated p values (*p < 0.05, **p < 0.01, ***p < 0.001, and ****p < 0.0001). Error bars were plotted as standard deviation or standard error of the mean as indicated in figure legends. For each comparison, data were presentative of at least three biological replicates. Sample sizes for each figure were given in the figure legends.

Sample sizes were determined based on variance from previous experiments for ABRs, DPOAE, and whole-cell patch-clamp recording. For other experiments, sample sizes were estimated base on variance obtained from pilot experiments.

## Acknowledgements

This work was supported by the grants from the National Natural Science Foundation of China (NSFC) (82000975), and the Fundamental Research Program Funding of Ninth People's Hospital Affiliated to Shanghai Jiao Tong University School of Medicine (JYZZ057) to TTD; and Shanghai Key Laboratory of Translational Medicine on Ear and Nose Diseases (14DZ2260300), Shanghai Municipal Science and Technology Major Project (21JC1404000), and the National Natural Science Foundation of China (NSFC) (81970872) to HW.

## Additional information

### Funding

| Funder | Grant reference number | Author |
| --- | --- | --- |
| National Natural Science Foundation of China | 82000975 | Ting-Ting Du |
| School of Medicine, Shanghai Jiao Tong University | JYZZ057 | Ting-Ting Du |
| Shanghai Key Laboratory of Translational Medicine on Ear and Nose Diseases | 14DZ2260300 | Hao Wu |
| Science and Technology Commission of Shanghai Municipality | 21JC1404000 | Hao Wu |
| National Natural Science Foundation of China | 81970872 | Hao Wu |

The funders had no role in study design, data collection, and interpretation, or the decision to submit the work for publication.

## Author contributions

Ling-Yun Zhou, performed experiments, analyzed data and wrote the manuscript; Chen-Xi Jin, performed AAV injections; Wen-Xiao Wang, Writing - review and editing, assisted with experiments and edited the manuscript; Lei Song, designed and supervised on the Whole-cell patch-clamp recording, and edited the manuscript; Jung-Bum Shin, initiated the project and edited the manuscript; Ting-Ting Du, Funding acquisition, conceived the project, analyzed the data, supervised the research and wrote the manuscript; Hao Wu, Funding acquisition, supervised the research and wrote the manuscript

## Author ORCIDs

Ling-Yun Zhou (iD) http://orcid.org/0009-0001-7041-4730
Jung-Bum Shin (iD) http://orcid.org/0000-0003-3047-0874
Ting-Ting Du (iD) https://orcid.org/0000-0003-0333-8739
Hao Wu (iD) https://orcid.org/0000-0002-5317-902X

## Ethics

The study protocol was approved by the Ethics Committee of Shanghai Jiao Tong University School of Medicine Affiliated Ninth People's Hospital (Shanghai, China) (SH9H-2020-A682-1). All animal maintenance and experimental procedures were performed by the recommendations in the Guide for the Institutional Animal Care and Use Committee (IACUC) of Shanghai Jiao Tong University.

## Decision letter and Author response

Decision letter https://doi.org/10.7554/eLife.90155.sa1
Author response https://doi.org/10.7554/eLife.90155.sa2

# Additional files

## Supplementary files

• Supplementary file 1. Primer sequences used for PCR and RT-qPCR.
• MDAR checklist

## Data availability

The raw sequence data reported in this paper have been deposited in the Genome Sequence Archive (*Chen et al., 2021*) in National Genomics Data Center (*Xue et al., 2022*), China National Center for Bioinformation/Beijing Institute of Genomics, Chinese Academy of Sciences (GSA: CRA010747), publicly accessible at https://ngdc.cncb.ac.cn/gsa/browse/CRA010747. All data generated or analyzed during this study are included in the manuscript and *Supplementary file 1*.

The following dataset was generated:

| Author(s) | Year | Dataset title | Dataset URL | Database and Identifier |
|---|---|---|---|---|
| Zhou L-Y, Jin C-X, Wang W-X, Song L, Shin J-B, T-T Du, Wu H | 2023 | Srf Mrtfb hair cell RNA-seq | https://ngdc.cncb.ac.cn/gsa/browse/CRA010747 | Genome Sequence Archive, CRA010747 |

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
