## [Editor Report]

This important study provides convincing evidence implicating two transcription factors in the regulation of the actin cytoskeleton that shapes the mechanosensory hair bundles of the inner ear's hair cells. Although the mechanistic understanding of their operation remains incomplete, the work will be of interest to biologists interested in the development and maintenance of the hair bundle.

---

## [Decision Letter]

**Decision letter after peer review:**

Thank you for submitting your article "Differential regulation of hair cell actin cytoskeleton mediated by SRF and MRTFB" for consideration by *eLife*. Your article has been reviewed by 2 peer reviewers, and the evaluation has been overseen by a Reviewing Editor and Jonathan Cooper as the Senior Editor.

Essential revisions:

1. The authors are expected to provide a mechanistic interpretation of the effects of Srf and Mrtfb deficiencies on the actin cytoskeleton of the hair bundle and cuticular plate. Without a testable working hypothesis, the takeaway of the work for a better understanding of hair-bundle development and maintenance remains unclear.

2. The reviewers have both expressed some concerns about data analysis based on fluorescent staining and provided a number of suggestions. They ought to be all addressed with care in the revised manuscript.

3. Adding TEM data to discuss F-actin organization would be a strong asset, although we recognize those experiments may be difficult to perform within a few months.

*Reviewer #1 (Recommendations for the authors):*

This manuscript provides a lot of data of relatively high quality. Yet, my major question after reading it: "And what?" These transcription factors are expected to affect hair cell formation, anyway. In fact, the effect of their deletion is surprisingly minor, suggesting that they are not the major players in the regulation of the hair cell cytoskeleton. It would be interesting to find a mechanism that would explain exactly how Srf and Mrtfb contribute to the stereocilia formation. Obviously, it is hard to talk about a specific mechanism for a transcription factor that regulates hundreds of genes. Yet, the effects of Srf and Mrtfb on the distribution of essential proteins between the stereocilia rows in the same bundle seems to be fundamentally different (Figure 2 vs Figure 4). Perhaps, quantification of these between-the-row differences following Barr-Gillespie's group is a "low-hanging" fruit that may bring you to a mechanistic model.

More specific comments:

Line 112: Correct to "…became undetectable at P0 in hair cells but not in supporting cells".

Figure 1G,H-M: It would be helpful to make these panels self-explanatory by adding subtitles like "fluorescence-based" or "SEM-based" measurements. Also, clear labeling of the age of the cells on SEM images and quantification panels J and M would be helpful.

Figure 7: The whole Panel F is confusing since it does not show any stereocilia tip localization of CNN2. Likewise, the bottom images in panel E are also confusing. Please, correct the figure legend to state clearly that ALL images in panel F were taken at the level of the cuticular plate and the bottom images in panel E were also taken at the level below stereocilia.

Line 521: Please, describe the details of "thiocarbohydrazide-osmium protocol", many labs use different versions of this protocol.

Literature cited in the Methods section seems to be missing in the list of references.

*Reviewer #2 (Recommendations for the authors):*

Line 79. Change to "…severing protein gelsolin…"

Line 108. Maybe due to expression in the brain too.

Line 114. Figure 6 reference out of order.

Line 116. How was the cuticular plate phalloidin labeling done? Was the whole CP volume integrated or was an x-y slice used?

Line 118. What do you mean by "abnormal distribution"?

Line 119. In Figure 1 and other figures-some of the lettering is too small.

Line 129. I don't think "curly" is a particularly good descriptive word for this phenomenon.

Line 141. Please label J and M with age and location.

Line 151. Kinocilia is plural, so "kinocilium" is the proper word to use in this sentence.

Line 161. Remove "The" before ESPN1.

Line 192. Mostly reduced? This result suggests that the antibodies are not highly specific.

Line 198. What is the "Mrtfa group"?

Line 221. While statistically significant, the size of the change is probably not biologically significant.

Line 283. Display all genes examined in both Figure 6C and Figure 6D.

Line 287. Saying "…the MRTFB-SRF transcription axis may not exist…" is unclear.

Line 324. Change "…calponin proteins that belong…" to "…calponin proteins, which belong…"

Line 376. Why was it surprising that length and width changes were observed in Srf and Mrtfb cKOs? I thought that was what the hypothesis was.

Line 397. Change "…yielded Srf-like…" to "…yielded a Srf-like…"

Line 398. The phrase "…lack of Srf or Mrtfb solo…" is confusing.

[Editors’ note: further revisions were suggested prior to acceptance, as described below.]

Thank you for resubmitting your work entitled "Differential regulation of hair cell actin cytoskeleton mediated by SRF and MRTFB" for further consideration by *eLife*. Your revised article has been evaluated by Jonathan Cooper (Senior Editor) and a Reviewing Editor.

The manuscript has been improved but there are some remaining issues that need to be addressed, as outlined below:

After discussion, both reviewers have agreed that a quantification of stereocilia dimensions in wild-type mice injected with Anc80L65-Cnn2 should be added to the manuscript. Although the manuscript does not make this point very clear, the data suggest that the level of Cnn2 expression may control actin polymerization in the hair bundle: Cnn2 levels are reduced in Srf cKO mice and injection of Anc80L65-Cnn2 in these mice restores stereocilia morphology and integrity of the cuticular plate (Figure 9). Following this logic, one would expect that overexpression of Cnn2 in wildtype hair cells could also promote elongation and widening of the stereocilia and more actin in the cuticular plate (unless elongation is already saturated), reinforcing the conclusion that Cnn2 is indeed an effector of Srf. In any case, this quantification would help develop the mechanistic discussion about the putative role of Srf.

Note that the new Figure 10 does not appear to be referenced and discussed in the main text.

Although less critical, the 3D-reconstruction data of the cuticular plate in IHCs could be added to the manuscript.

*Reviewer #1 (Recommendations for the authors):*

This revised version of the manuscript has very welcome additions of a more thorough quantitative analysis of immunolabeling, new electron microscopy data, a better description of quantitative analysis, and a slightly revised discussion of the potential role of Srf and Mrtfb in the formation and/or maintenance of stereocilia bundles. The revision answered most of my previous comments, except for hypothetical molecular mechanisms of Srf and Mrtfb function, which may be hard to address. However, there is one essential issue that I've noticed after reading the revised manuscript.

The argument that CNN2 partially restores Srf cKO phenotype is based only on the comparison of the wildtype hair cells transfected with Anc80L65-GFP and the Srf cKO hair cells transfected with Anc80L65-GFP or Anc80L65-Cnn2 (Figure 9H,J and Suppl Figure 9G). What if an overexpression of Cnn2 forces stereocilia to grow even in the normal, wild-type hair cells? Then the observed stereocilia growth in Scf cKO mice after Anc80L65-Cnn2 transfection may be completely independent of Srf function. Therefore, the data analysis in Figure 9H,J and Suppl Figure 9G is missing one important control – wildtype hair cells transfected with Anc80L65-Cnn2. Based on a single image in Figure 9E, the authors did perform these experiments but, unfortunately, did not quantify stereocilia dimensions.

I also do not understand why a powerful quantification of the volume of cuticular plates with Imaris software was provided only in the letter to the reviewers but not included in the manuscript figures.

---

## [Author Response]

Essential revisions:1. The authors are expected to provide a mechanistic interpretation of the effects of Srf and Mrtfb deficiencies on the actin cytoskeleton of the hair bundle and cuticular plate. Without a testable working hypothesis, the takeaway of the work for a better understanding of hair-bundle development and maintenance remains unclear.2. The reviewers have both expressed some concerns about data analysis based on fluorescent staining and provided a number of suggestions. They ought to be all addressed with care in the revised manuscript.3. Adding TEM data to discuss F-actin organization would be a strong asset, although we recognize those experiments may be difficult to perform within a few months.

We would like to thank the reviewers for the opportunity to address the issues raised in their comments. We have conducted additional experiments to address most of the concerns. In the revised manuscript, we have put more efforts on (1) interpreting the mechanistic understanding of the SRF and MRTFB in the hair cell (the revised Figures 3, 6 and 10), (2) describing the procedure of image quantification and analyses (the Materials and methods-Immunocytochemistry) and (3) conducting TEM analysis to examine directly the defects of F-actin organization of the cuticular plate in the mutants (the revised Figures 1 and 4). In the revised version, changes to our manuscript were all highlighted by using red colored text in the paper. Please find below a point-by-point address of the critique points.

Reviewer #1 (Recommendations for the authors):This manuscript provides a lot of data of relatively high quality. Yet, my major question after reading it: "And what?" These transcription factors are expected to affect hair cell formation, anyway. In fact, the effect of their deletion is surprisingly minor, suggesting that they are not the major players in the regulation of the hair cell cytoskeleton. It would be interesting to find a mechanism that would explain exactly how Srf and Mrtfb contribute to the stereocilia formation. Obviously, it is hard to talk about a specific mechanism for a transcription factor that regulates hundreds of genes. Yet, the effects of Srf and Mrtfb on the distribution of essential proteins between the stereocilia rows in the same bundle seems to be fundamentally different (Figure 2 vs Figure 4). Perhaps, quantification of these between-the-row differences following Barr-Gillespie's group is a "low-hanging" fruit that may bring you to a mechanistic model.

As the reviewer mentioned, it is difficult to figure out which gene is the key player among hundreds of downstream targets of SRF or MRTFB. Our efforts in this revised manuscript focused on indepth comparisons of stereocilia tip proteins’ localization pattern among control, *Srf* cKO and *Mrtfb* cKO during stereocilia development. Following the reviewer’s suggestion, we performed ESPN1, EPS8 and GNAI3 staining on the control and the mutant’s hair cells at P4, P10 and P15. The expression level, the row-specific distribution and the irregular labeling of tip proteins were analyzed and plotted in the revised Figure 3 and Figure 6. The results indicated that during stereocilia development, the effects of SRF and MRTFB on tip proteins’ localization patterns were fundamentally different, which were highly consistent with the severity of the morphological defects of stereocilia dimensions in *Srf* cKO and *Mrtfb* cKO hair cells.

We summarized the functional consequences among the control, *Srf* cKO and *Mrtfb* cKO mice in revised Figure 10 as working model, including the localization pattern of tip proteins during stereocilia development, represented by EPS8, as well as the morphological defects of stereocilia dimensions and cuticular plate.

We agree with the reviewer’s surprise that late embryonic deletion of SRF, supposedly a master regulator of the actin cytoskeletal network, had such mild effects on the development of the hair cell, a cell type with uniquely prominent F-actin-based structures. Instead, our data suggests that SRF is more important for the maintenance of the hair cell’s F-actin structures, based on the rapid postnatal degeneration of the stereocilia and cuticular plate. We believe that this finding is important, as it will provide the foundation for future research on the role of SRF and its downstream components in the long-term maintenance of the hair bundle, with potential implications for understanding age-related hearing loss. We believe that this is a reasonable and meaningful extrapolation of our data, thus included a sentence elaborating on this notion at the end of the Discussion section.

More specific comments:Line 112: Correct to "…became undetectable at P0 in hair cells but not in supporting cells".

This sentence has been corrected.

Figure 1G,H-M: It would be helpful to make these panels self-explanatory by adding subtitles like "fluorescence-based" or "SEM-based" measurements. Also, clear labeling of the age of the cells on SEM images and quantification panels J and M would be helpful.

Thanks for the suggestion. The subtitles and the labeling of ages have been modified in the revised Figure 2 and Figure 5.

Figure 7: The whole Panel F is confusing since it does not show any stereocilia tip localization of CNN2. Likewise, the bottom images in panel E are also confusing. Please, correct the figure legend to state clearly that ALL images in panel F were taken at the level of the cuticular plate and the bottom images in panel E were also taken at the level below stereocilia.

We agree with the reviewer. We have corrected the figure legend to state the images more clearly, please see the revised Figure 9 legend.

Line 521: Please, describe the details of "thiocarbohydrazide-osmium protocol", many labs use different versions of this protocol.

We have added more details of thiocarbohydrazide-osmium protocol in this section.

Literature cited in the Methods section seems to be missing in the list of references.

We have included the literature cited in the methods in the list of references.

Reviewer #2 (Recommendations for the authors):Line 79. Change to "…severing protein gelsolin…"

Yes, we have changed the sentence as suggested.

Line 108. Maybe due to expression in the brain too.

Yes, we agree. As described in Siegfried Alberti’s study, the forebrain-specific deletion of *Srf* in mice causes severe abnormalities, including balance impairments, lack of interest in feeding and reduced body weight. *Srf ^fl/fl^; CamKIIa-iCre* mice usually died around P20. We have made changes to the causes of the weak appearance of *Srf* cKO mice and added this paper below in the reference list.

Reference:

Alberti, S. et al. Neuronal migration in the murine rostral migratory stream requires serum response factor. Proc. Natl. Acad. Sci. USA 102, 6148–6153 (2005).

Line 114. Figure 6 reference out of order.

This sentence now refers to Figure 1—figure supplement 1E.

Line 116. How was the cuticular plate phalloidin labeling done? Was the whole CP volume integrated or was an x-y slice used?

In Figures 1A and 1B, the quantification of phalloidin intensity was processed using ImageJ. We first flattened these images to 2-D images by applying maximum intensity projection along the zaxis of the whole cuticular plate, then circled the area of the cuticular plate labeled with phalloidin and quantified the intensity. We have added more detail on the quantification of the cuticular plate phalloidin intensity to the “Materials and methods- Immunocytochemistry” section.

Line 118. What do you mean by "abnormal distribution"?

We have changed this sentence to “displayed a more punctate distribution in the cuticular plate”.

Line 119. In Figure 1 and other figures-some of the lettering is too small.

We have gone through all the figures and made appropriate changes to the size of the lettering.

Line 129. I don't think "curly" is a particularly good descriptive word for this phenomenon.

We have changed the sentence to “The deformation of the overlying plasma membrane above the cuticular plate of OHCs were also revealed by scanning electron microscopy (SEM) in mutants”.

Line 141. Please label J and M with age and location.

Yes, we have added age and location to the revised Figure 2 and Figure 5.

Line 151. Kinocilia is plural, so "kinocilium" is the proper word to use in this sentence.

We have corrected it to “kinocilium”.

Line 161. Remove "The" before ESPN1.

We have removed “The”.

Line 192. Mostly reduced? This result suggests that the antibodies are not highly specific.

Yes, we agree with the reviewer. We have tried a few other commercial MRTFA and MRTFB antibodies, but none of them showed good specificity in the cochlea. We have decided to change the word “mostly” to “partially” to make the conclusion more accurate.

Line 198. What is the "Mrtfa group"?

We have changed the “*Mrtfa* group” to “*Mrtfa* cKO mice”.

Line 221. While statistically significant, the size of the change is probably not biologically significant.

Yes, we agree. After performing ESPN1, EPS8 and GNAI3 staining at different ages during stereocilia development, the results overall suggested that the biologically significant reduction of relative row 1 tip fluorescence signal of EPS8 and GNAI3 appeared in *Mrtfb* cKO at P15.

Line 283. Display all genes examined in both Figure 6C and Figure 6D.

Yes, we have included all the genes examined by RT-qPCR in the revised Figures 8C and 8D in this sentence.

Line 287. Saying "…the MRTFB-SRF transcription axis may not exist…" is unclear.

Yes, we agree. We have deleted this sentence and made a few changes as follows “none of the down-regulated genes in *Srf* targets and *Mrtfb* targets were common, highly suggesting that in hair cells, SRF or MRTFB may regulate the distinct profiles of genes that affect the development and function of the actin-based structures.”

Line 324. Change "…calponin proteins that belong…" to "…calponin proteins, which belong…"

We have corrected it.

Line 376. Why was it surprising that length and width changes were observed in Srf and Mrtfb cKOs? I thought that was what the hypothesis was.

Yes, we agree with the reviewer. We have made appropriate changes to this sentence.

Line 397. Change "…yielded Srf-like…" to "…yielded a Srf-like…"

We have changed “Srf-like” to “a Srf-like”.

Line 398. The phrase "…lack of Srf or Mrtfb solo…" is confusing.

Yes, we agree with the reviewer. We have made appropriate changes to this sentence.

[Editors’ note: what follows is the authors’ response to the second round of review.]

The manuscript has been improved but there are some remaining issues that need to be addressed, as outlined below:After discussion, both reviewers have agreed that a quantification of stereocilia dimensions in wild-type mice injected with Anc80L65-Cnn2 should be added to the manuscript. Although the manuscript does not make this point very clear, the data suggest that the level of Cnn2 expression may control actin polymerization in the hair bundle: Cnn2 levels are reduced in Srf cKO mice and injection of Anc80L65-Cnn2 in these mice restores stereocilia morphology and integrity of the cuticular plate (Figure 9). Following this logic, one would expect that overexpression of Cnn2 in wildtype hair cells could also promote elongation and widening of the stereocilia and more actin in the cuticular plate (unless elongation is already saturated), reinforcing the conclusion that Cnn2 is indeed an effector of Srf. In any case, this quantification would help develop the mechanistic discussion about the putative role of Srf.

Yes, we agree that the quantitative analysis of stereocilia morphology and integrity of the cuticular plate in the control mice injected with Anc80L65-Cnn2 is essential and helpful for understanding the role of Srf and Cnn2. The quantifications and representative images were showed in this new revised Figure 9G-J and Figure 9—figure supplement 1F,G.

Note that the new Figure 10 does not appear to be referenced and discussed in the main text.

We apologize for not mentioning the Figure 10 in the paper. Now the revised Figure 10 has been referenced and discussed in the first paragraph of the Discussion section.

Although less critical, the 3D-reconstruction data of the cuticular plate in IHCs could be added to the manuscript.

We have added the 3D-reconstruction data of the cuticular plate in IHCs in this new revised Figure 1F,G and Figure 4E,F.

Reviewer #1 (Recommendations for the authors):This revised version of the manuscript has very welcome additions of a more thorough quantitative analysis of immunolabeling, new electron microscopy data, a better description of quantitative analysis, and a slightly revised discussion of the potential role of Srf and Mrtfb in the formation and/or maintenance of stereocilia bundles. The revision answered most of my previous comments, except for hypothetical molecular mechanisms of Srf and Mrtfb function, which may be hard to address. However, there is one essential issue that I've noticed after reading the revised manuscript.The argument that CNN2 partially restores Srf cKO phenotype is based only on the comparison of the wildtype hair cells transfected with Anc80L65-GFP and the Srf cKO hair cells transfected with Anc80L65-GFP or Anc80L65-Cnn2 (Figure 9H,J and Suppl Figure 9G). What if an overexpression of Cnn2 forces stereocilia to grow even in the normal, wild-type hair cells? Then the observed stereocilia growth in Scf cKO mice after Anc80L65-Cnn2 transfection may be completely independent of Srf function. Therefore, the data analysis in Figure 9H,J and Suppl Figure 9G is missing one important control – wildtype hair cells transfected with Anc80L65-Cnn2. Based on a single image in Figure 9E, the authors did perform these experiments but, unfortunately, did not quantify stereocilia dimensions.

Thanks for the reviewer’s suggestion. Now we have added the quantifications, including the relative F-actin intensity change of the cuticular plate, row 1 stereocilia length and width of hair cells, and representative images in the new revised Figure 9G-J and Figure 9—figure supplement 1F,G. The quantification results suggested that Cnn2 may function in regulating stereocilia actin polymerization and the F-actin organization of the cuticular plate and the precise expression level of Cnn2 controlled by Srf is essential for the shaping of stereocilia architecture.

Based on these above results, we agree with the reviewer, and we couldn’t exclude the possibility that “the observed stereocilia growth in Scf cKO mice after Anc80L65-Cnn2 transfection may be completely independent of Srf function”. Further investigation is necessary to validate the direct transcription factor (Srf)-target gene (Cnn2) binding in isolated hair cells using tests like ChIP-PCR, which we were not able to finish in this revised version due to time constraints.

I also do not understand why a powerful quantification of the volume of cuticular plates with Imaris software was provided only in the letter to the reviewers but not included in the manuscript figures.

We have added the 3D-reconstruction analysis data of the cuticular plate volume using Imaris in this new revised Figure 1F,G and Figure 4E,F.